

# Evaluating primary productivity, ripple effect and resilience of fluvial ecosystems: a new approach to assessing environmental flow requirement

Yui Shinozaki[1] and Naoki Shirakawa[1]

[1]Graduate School of Systems and Information Engineering, University of Tsukuba, Tennoudai 1-1-1 Tsukuba, 3058573, Japan

*Correspondence to*: Yui Shinozaki (s1630197@u.tsukuba.ac.jp)

## Abstract

Productivity, ripple effect and resilience are characteristics of fluvial ecosystems. To evaluate these factors and develop new criteria for estimating environmental flow requirement (EFR), we propose a fluvial biomass model which calculates the amount of aquatic biomass accumulated through physical and climatic processes. Using this model, we introduce the indices of Contribution to Downstream Ecosystems (CDE) and Ecological Recovery Time (ERT) and apply them in the estimation of global EFRs. Average EFR values were estimated at 42% of mean annual discharge. In comparison with previous global EFR estimates based on flow regime only, our model suggests 20%–50% higher values in monsoonal and savanna regions with high ecological richness, and in the lower reaches of large rivers in the higher latitudes where primary productivity is low and ecosystems largely depend on allochthonous biomass supply. The main advantage of our model is the ability to set variable EFRs within a river basin based on differences in ecological characteristics driven by climatic conditions and tributaries. Taking such longitudinal differences in EFR into account aids in developing integrated water allocation strategies by reflecting differences in water resource availability for humans within a catchment.

## 1 Introduction

Substantial amounts of water are globally being withdrawn from rivers in order to supply human needs. More than 800 000 dams have been built since the beginning of the twentieth century, resulting in the fragmentation of 75% of major rivers (Richter, 2006; Biemans et al., 2011). The total cumulative storage of large dams is about 20% of global runoff (Vörösmarty et al., 1997). Reservoirs secure a steady water supply where water availability is highly susceptible to seasonal fluctuations and enable the affected regions to achieve socio-economic development. On the other hand, flow regulations by dams and other water intake structures alter natural flow regimes, such as the timing and magnitude of downstream flow of water,





nutrients and sediments, which can result in serious degradations of freshwater ecosystems (Renöfält et al., 2010; Poff, 2010; Vörösmarty et al., 2010). Vörösmarty reported that about 65% of rivers worldwide are now considered under moderate to high threat in terms of water scarcity for human use and of biodiversity. Healthy rivers provide not only water resources but also a variety of ecosystem services indispensable for humans, and ecological degradation associated with a reduction in

flow decreases opportunities to make use of these services. Sustainable freshwater resource management thus requires the protection of entire ecosystems.

To sustain a healthy fluvial ecosystem, it is necessary to maintain certain amount of water in the channel. This idea has been developed into the concept of environmental flow requirement (EFR). EFR is defined as the quantity, timing and quality of water flow required for sustaining freshwater and estuarine ecosystems and the human livelihoods and well-being

that depend on these ecosystems (Brisbane Declaration of 2007). How much environmental flow should be allocated within a channel is now an issue of global concern. More than 200 methods for estimating EFRs have been proposed in the past decades (Tharme, 2003). These are classified into four major classes (Pastor at al., 2014): hydrological methods, hydraulic methods, habitat simulation methods and holistic methods. Early approaches aimed to define either minimum or average flows to support key fish species or maintain riverine habitat; however, these are now regarded as too simplistic to support

complex flow-dependent ecosystem functions (Acreman and Ferguson, 2010; Shafroth et al.; 2010, Pahl-Wostl et al., 2013). It is now generally recognized that EFRs should vary in space and time to sustain the desired ecosystem state, together with the bundles of ecosystem services. The ELOHA framework (Poff, 2010), focusing on ecological responses and flow regime, is an empirically testable framework based in ecological science and employing existing ecohydrological knowledge and hydrology modelling tools. This approach enables the development of flow management guidelines for ecological

sustainability in a target region.

At a global scale, the concept of EFR has been found useful in many localities, particularly in developing regions of the world that are under increasing pressure to develop their water resources (Smakhtin et al., 2006). Recent applications of EFR have been reported from Vietnam (Babel et al., 2011), China (Yang and Yang, 2014), Brazil (Castro at al., 2015), Sri Lanka (Eriyagama et al., 2016), Turkey (Karakoyun et al., 2015), Bangladesh (Akter and Ali, 2012), Nepal (Smakhtin et al.,

2006.; Rijal and Alfredsen, 2015) and Iran (Nia et al., 2016). Water resources are now frequently being exchanged across rivers and catchments, and there may also be longitudinal disproportionalities in water resource availability. In order to assess where enough water is available for withdrawals and meeting other human demands, it is necessary to estimate how much water is needed to sustain freshwater ecosystems at a global scale (Pastor et al., 2014). Several global EFR models have been proposed to this end. As a first attempt, Smakhtin et al. (2004) proposed a pilot global model based on a

hydrological method. In this model, EFR is a combination of low-flow requirement (LFR), which is necessary for aquatic organisms to survive, and high-flow requirement (HFR), which is required for maintenance of channels and natural morphology. Four levels of EFRs are defined according to environmental status: *natural* (unmodified), *good* (slightly to moderately modified), *fair* (moderately to considerably modified) and *poor* (critically modified to degraded). For *natural* status, LFR is represented by the flow level which is exceeded 75 percent of the time (Q75). Similarly, LFR is defined as




Q50 for *good* status and Q90 for *fair* status. *Poor* ecosystem conditions are not considered acceptable from a management perspective (Smakhtin et al. 2004) and are thus not allocated an EFR. HFR is allocated as a percentage of mean annual discharge for each status.

Hanasaki et al. (2008) estimated global EFR as a percentage of mean annual discharge depending on the climatic classification of each region. Conceptually, EFRs have shifted from a means of protecting environmental needs in a water resource planning context to a management cycle for sustainable management of social-ecological aquatic systems for multiple benefits and scales (Matthews et al., 2014). However, in contrast to the regional case studies, global EFR methods are still mainly based on simple hydrological methods, forced by a lack of global ecohydrological data to limit themselves to mean annual discharge and some flow variabilities (Richter et al., 2006). Stream flow has often been treated as the "master variable" since it can be readily described by indices (Poff and Matthews, 2013). Aside from ecohydrological factors such as stream organisms, biomass, sediment and nutrient transportation, which cannot be directly controlled by managers of water infrastructures, the environmental flow concept is based on the principle that a) hydrologic alteration impairs ecosystem functions and b) hydrologic indices of alteration can be used as proxies for this. However, focusing merely on hydrological valuables would appear to be insufficient when considering whole-ecosystem services. Yoshikawa et al. (2014) pointed out that variables of previous global models were estimated without considering explicit linkages to freshwater ecosystem structures and functions. They proposed a novel global scale approach using the ESR-FLAVAR method to focus on the relationships between species richness of fish and some characteristics of river discharge. This method combines the concept of FSR-MAD (Xenopoulos et al., 2005), which establishes a relationship between fish species richness and mean annual discharge, and the method of Iwasaki et al. (2012). This type of approach offers a new perspective on determining EFR, and on global modelling with a focus on ecological diversity. However, fish species richness can only be used in evaluating a single aspect of the totality of ecosystem services. To use EFR as an effective tool for integrated water resource management, a method is required that enables the evaluation of ecological richness as a whole. Such a method would offer global perspectives and allow comparisons of ecosystem services between regions, or the determination of particularly vulnerable riverine biodiversity hotspots in need of preferential protection. This kind of information may also guide global conservation efforts and provide inform what kinds of management interventions of infrastructure design are needed (Poff and Matthews, 2013).

In the context of fluvial ecosystems as a whole, ecosystem deterioration due to decreased river discharge may be described by three characteristics: deterioration of productivity, resilience and ripple effects. For example, dam construction may directly alter the natural habitat on the site and affect the primary production and quantitative balance of aquatic life (*productivity*). It also has effects on the natural flow regime, which extend to downstream (sometimes including estuarine and marine) ecosystems (*ripple effect*). In addition, decrease in river discharge weakens the resilience of ecosystems and render them more vulnerable to small changes in external forces and disturbances (*resilienc*e). These descriptors can be connected to widely known concepts used to describe riverine ecosystems from a material and energy flux perspective: The Riverine Productivity Model (RPM; Thorp et al., 1994), the River Continuum Concept (RCC; Vannote, 1980) and the Flood



Pulse Concept (FPC; Junk et al., 1989). RPM emphasizes domestic productivity; in RCC, downstream transportation of material and energy are important to sustain fluvial ecosystems; and FPC focuses on the importance of flooding disturbance in creating the basis of aquatic habitats.

By introducing indices for the above three characteristics, we believe it is possible to evaluate the regional differences in ecosystem quality that need to be distinguished when setting EFR. Still, no current EFR method covers all of these characteristics. Among existing methods, habitat simulation methods focus on the quality of habitat for aquatic life. They aim to quantify the suitability of physical habitat for target species under different flow regimes. IFIM (Tharme, 2003) and PHABSIM; Maddock, 1999) are widely known. These methods, however, focus on only a single or a few target species and their habitat within a specific section of river and are thus not able to evaluate productivity and spatial patterns of fluvial

ecosystems.

To evaluate *productivity*, *ripple effect* and *resilience* in fluvial ecosystems, we propose a conceptual model focused on plant biomass. This metric was chosen because plants are dominant component of biomass which are directly controlling primary productivity and nutrient retention (Fox, 2004), and thus, plant biomass serves as a basis for sustaining fluvial ecosystems. Instead of offering specific values and criteria, in this paper, we provide a perspective on the conceptual method

for setting EFRs. Our fluvial biomass model can express the *productivity* of plants in rivers by considering net primary productivity of aquatic plants, biomass accumulation through the process of longitudinal transportation, supply from terrestrial vegetation, and decomposition/mineralisation. Based on this model, we introduce two indices, Contribution to Downstream Ecosystems (CDE) and Ecological Recovery Time (ERT) to evaluate *ripple effect* and *resilience*, respectively. We then these use these ecological characteristics to improve an existing EFR method based on the widely used hydrological

method proposed by Tennant (1976) and estimate EFR at a global scale. Finally, we discuss the advantages of our new concept under a sustainable water resource management perspective by comparison with an existing global model.

## 2 Method

### 2.1 Model Concept

#### 2.1.1 Evaluation of essential characteristics of fluvial ecosystems

We propose a conceptual model to evaluate the *productivity*, *ripple effect* and *resilience* of fluvial ecosystems. A fluvial ecosystem is characterized by energy and material transportation within and exchange outside the system. Three well-known concepts which can be used to explain fluvial ecosystems from a material and energy flux perspective are the River Continuum Concept (RCC), the Flood Pulse Concept (FPC) and the Riverine Productivity Model (RPM). RCC emphasizes the role of longitudinal transportation of energy and material in characterizing the biota of fluvial ecosystems by explaining

the sequence of individual sections of a river as determined by physical characteristics such as velocity, water depth, width, turbulence and terrestrial canopies. FPC focuses on the role of the exchange of energy and materials between river and flood



plain during flood events, as this creates the basic riverine environment during non-flooding periods. While RCC and FPC emphasize the importance of interactions with spaces outside the system in characterizing the ecosystems of a specific aquatic system, PRC incorporates domestic production as an essential factor for understanding the characteristics of fluvial ecosystems. PRC in particular can be applied to large continental rivers where the retention time of water is long (Thorp et al., 1994). While all three concepts are qualitative, they are very helpful in gaining an understanding of the fundamentals of fluvial ecosystems through the process of material and energy flux. Although it may be infeasible to expect a representation of all physical and biological responses (which may be highly complex and contain unknown mechanisms), many approaches have been proposed to describe this type of flux in and around riverine ecosystems. In this paper, we focus on the biomass budget of primary producers (plants). We developed a fluvial biomass model describing the biomass budget within a river channel (Figure 1). We define aquatic plant (autochthonous) biomass as vegetation biomass growing in water, either rooted to riverbed or substrate, or freely floating (including algae). Riparian plants that are irregularly submerged during floods, as well as terrestrial woody debris and litter falls, make up allochthonous biomass.

This model expresses primary production and biomass accumulation of plants in rivers by considering net primary productivity (NPP) of aquatic plants, longitudinal biomass flow within a channel and biomass supply from terrestrial vegetation. In addition to these processes, a certain amount of biomass leaves the system through decomposition and mineralisation. Theoretically, the processes occur simultaneously and the amount of biomass is kept in equilibrium. Biomass is calculated with the following formula:

$$\partial B = Bu + C + NPP - F - R \qquad (1)$$
$$C = \beta B \qquad (2)$$
$$F = f\left(\frac{B}{\alpha \Delta x}V\right) \qquad (3)$$
$$R = rB \qquad (4)$$

Here B is biomass (g), $B_u$ is allochthonous biomass produced upstream (g), C is allochthonous biomass provided from terrestrial vegetation, NPP is net primary production within a given period of time (g), F is biomass flow out of the area, R is biomass dissipation, V is flow velocity for a given period of time (m), α is coefficient of meander, and $\Delta x$ is river length (m). The following parameters are given: $\beta$ is the proportion of C in fluvial biomass, $f$ is flux rate, and $r$ is the rate of decomposition and mineralization.

### 2.1.2 Evaluation of *productivity*

The importance of domestic productivity for sustaining fluvial ecosystems is supported by the RPM of Thorp (1994). NPP is a fundamental factor in defining how much biomass accumulates within a channel and what amount of biomass affects the structures of food webs and trophic levels. Gross primary productivity (GPP) is the rate at which primary producers store



energy through photosynthesis, a part of which is consumed for respiration and the remainder accumulated as biomass; the rate describing the latter process is referred to as NPP. A number of stream ecology studies have investigated ecological diversity and productivity in rivers (Power and Dietrich, 2002; Aoki, 2003; Bunn et al., 2006; Hunt et al., 2011). Aoki and Mizushima (2001) demonstrated quantitatively that an increase in biomass diversity across all trophic levels leads to an

increase in systemic stability in aquatic ecosystems. To evaluate productivity within a channel, we used the fluvial biomass model to calculate the amount of plant biomass.

### 2.1.3 Evaluation of *ripple effect*

How much plant biomass flows downstream in a river is a key indicator to evaluate *ripple effect*. The importance of longitudinal transportation of materials and energy is supported by RCC as developed by Vannote (1980). This concept is

based on the idea that a river is an open ecosystem with a constant energy flow from source to mouth, and the physical parameters as well as biological factors in a river are maintained as a dynamic equilibrium. In order to evaluate ripple effect, it is necessary to understand that the longitudinal continuity also relates to the autonomy of aquatic ecosystems. Autonomy of plant biomass is evaluated by determining the origin of plant biomass which a consumer organism needs to survive, both autochthonous and allochthonous. Many case studies have investigated primary productivity and the origins of plant biomass

in lotic ecosystems (e.g., Fisher et al., 1973; Mulholland, 1981; Webster and Meyer, 1997). The rate of autonomy varies between rivers, sections of rivers, season, turbidity and rate of disturbances (Bunn et al., 2006). Zeug et al. (2008) reported that terrestrial carbon sources may significantly contribute to consumer biomass in large rivers. In general, the majority of plant biomass in a mountainous stream is allochthonous, where the watercourse is relatively narrow and there is a dense canopy cover above the river. In such stream sections, most plant biomass is provided as litter by terrestrial forest, because

sunlight does not reach the water body and primary productivity of algae and riparian vegetation is low. In contrast, in the middle and lower parts of a stream where it is wider, flow is relatively moderate and shallow, and adequate sunlight penetrates to the water, primary production tends to be more autochthonous (Odum, 2004). In dryland rivers, gross primary production is one to two orders of magnitude greater than in temperate rivers; this is attributable to high light intensity, low current velocity, high temperature and intensive internal recycling of nutrients. Here, allochthonous inputs may not be an

essential source of carbon for consumers (Velasco et al., 2003; Bunn, 2006). Where the ecosystem depends on allochthonous carbon sources, the river section is more vulnerable to environmental destruction of its biomass source. By evaluating ripple effect, it is possible to highlight these regions.

However, a majority of existing EFR methods focus only on a specific section of a river, or set a single criterion for the whole catchment without considering longitudinal transportation of organic matter. Taking *ripple effect* into account is

beneficial not only for gaining an understanding of the ecological functions in a river channel but also for developing effective water resource management. The river Nile provides a good example. Primary productivity in mid- to downstream sections is low because they are located in desert zones, yet these areas have been cultivated since antiquity. This is made possible by fertile soil containing nutrients and organic matter having been transported there from upstream areas with





higher primary productivity. In the present study, we introduced CDE to incorporate *ripple effect*, and represented it by using the fluvial biomass model to calculate biomass transported downstream. Although many types of material are transported by flowing water, we focused on plant biomass because it provides food resources for consumer organisms.

### 2.1.4 Evaluation of *resilience*

*Resilience* describes the capacity of an ecosystem to recover equilibrium after disturbance. As a measure, it is helpful for identifying regions where vulnerability to disturbance is high. If an ecosystem is vulnerable, greater attention is required to conserve it in a healthy state. Although it does not express resilience directly, FPC offers a perspective which indicates that nonstationary events play an important role in supporting fluvial ecosystems during non-flooding periods. The time that plant biomass needs to recover to the original state after elimination (by either natural or artificial disturbances) may be a
helpful indicator of the resilience of a specific river section. It is known that some regions with an apparently abundant standing crop of vegetation take extremely long times to recover to the original state after destruction. For example, tropical rain forest contains about 3.5 times more plant biomass than temperate forest (Whittaker and Likens, 1973). However, because the top soil in some rain forests is only a thin and easily eroded layer, it takes an extremely long time to recover once destroyed. Such regions are highly vulnerable, and it is thus necessary to carefully maintain the natural status and keep
human impacts to a minimum. In contrast, some kinds of ecosystem recover quickly after disturbance. The Japanese landscape category of *Satoyama* provides a good example. *Satoyama* is a traditional agricultural landscape which contains a variety of land uses such as backyard forests, paddy fields, grasslands, ponds and streams. It contains higher species diversity than virgin forest because traditional human activities have maintained a combination of various habitats (Katoh et al., 2009). A monsoonal climate with high primary productivity supports a strong potential of ecological resilience, allowing humans to
make use of natural resources without exhausting the ecosystem. In the present paper, we introduce the concept of ERT to describe *resilience*.

### 2.2 Indices

### 2.2.1 Contribution to Downstream Ecosystems

To represent ripple effect, we introduced the concept of CDE, which is an index of the amount of biomass that a specific
river section contributes to the upkeep of downstream ecosystems. It is derived from the amount of biomass flowing down from the section, expressed as *F* in Formula (3). To allow comparisons between regions, CDE is normalized to the maximum value within the domain of analysis (range 0 to 1). A larger CDE corresponds to a higher contribution to downstream ecosystems, i.e., a larger ripple effect.



### 2.2.2 Ecological recovery time

To describe resilience, we introduced the concept of ERT, which is calculated using the fluvial biomass model. Plant biomass in the river ($B$) has an initial state of 0, expressing the situation that all biomass at the site has been removed due to disturbance. The calculation continues until $B$ recovers to a state of equilibrium. The number of time steps required is

referred to as ERT. Longer ERT indicates that the resilience of the site is low and that the ecosystem is vulnerable to disturbance.

In addition to the global scale calculation, two other scales of ecological destruction are considered: point destruction and catchment scale destruction. Point destruction represents events such as dredging or river modification along a limited section of a channel. In this case, biomass is supplied immediately from unaffected upstream and terrestrial areas.

Catchment scale destruction represents longitudinal deterioration caused by, e.g., dam construction upstream. As an example, ERT was evaluated at the three scales for a section of the Mekong. The target cell was located near Phnom Penh (11°33′N 104°55′E), the capital city of Cambodia, on the lower Mekong. For point destruction, biomass of the target cell was set to 0 as an initial state; for partial catchment scale destruction, this initial state was applied to the entire course up to the mid-reach (near Vientiane in Laos); for whole catchment scale destruction, biomass of the entire river was set to 0.

Ecological recovery time is expressed as the number of days, since the model has a daily time step. Note however that since the results presented in this paper are relative values meant to allow comparison of magnitude correlation between regions, it is not an objective of the model to express real recovery processes of fluvial ecosystems, because these are affected by numerous complex factors which cannot be reflected by the model.

### 2.2.3 Trophic level index

The productivity of fluvial ecosystems can be evaluated by calculating the amount of biomass (Sect. 2.1.2). Here, we attempted to convert biomass into a measure of trophic levels. Discrete quantities are more useful than continuous values in comparing different regions or determining threshold levels and target species for applications such as conservation strategies and setting EFRs. To this end, we introduced a trophic level index (TI). Although this index is not an essential part of our model, it constitutes a useful sub-model to clarify interpretation of the calculated fluvial biomass.

A conceptual illustration of TI is shown in Figure 2. The bottom trophic level is plant biomass. General target species are set for each trophic level. TI = 0 means that no vegetation-based ecosystem can exist. Target species for TI = 1 are plants as primary producers; for TI = 2, macroinvertebrate grazers; for TI = 3, small fish as primary predators; for TI = 4, mid-sized fish; and for TI = 5, apex predators such as large fish or birds. Energy transfer to higher trophic levels is incomplete due to losses from non-predatory death, respiration and excretion. When energy is transferred to the next trophic

level, ~10% of energy is fixed as biomass and the rest is used for metabolic processes (Odum 2004). The higher TI signifies that there is a highly-productive ecosystem which has more potential to contain various ecosystem services.



Parameters used to calculate TI are shown in Table 1. Each target species is assigned a basic daily metabolic rate ($\varphi$ kcal/g), an energy content when consumed by organisms of a higher trophic level ($e$ kcal/g), an energy transfer efficiency ($\delta$%), and a weight ($w$ g). Table 2 shows the process to calculate the number of trophic levels from a given plant biomass on a caloric basis. Applying the parameters of Table 1, energy necessary for an individual to survive (kcal/individual/year) and

energy content of an individual transferred to a predator of a higher trophic level (kcal/individual) are calculated. For example, whether the small fish on the third trophic level can survive depends on the number of macroinvertebrates which are supported by the available plant biomass. Let the basic metabolic rate of small fish be $\varphi_3$, energy content $e_3$, energy transfer efficiency $\delta_3$ and individual weight $w_3$. Then, energy necessary for an individual small fish to survive one year (365 days) can be described as $(365\varphi_3 w_3/\delta_3)$. Energy of an individual macroinvertebrate taken by small fish can be described as

$e_2 w_2$. Using this system, the number of macroinvertebrates necessary for a small fish to survive is $(365\varphi_3 w_3/\delta_3) \times (1/e_2 w_2)$. Similarly, the minimum amount of plants necessary for a macroinvertebrate to survive is $II = (365\varphi_2 w_2/\delta_2) \times (1/e_1)$. The minimum amount of plants necessary for a small fish to survive is the product of macroinvertebrate and small fish values: $III = \{(365\varphi_2 w_2/\delta_2) \times (1/e_1)\} \times \{(365\varphi_3 w_3/\delta_3) \times (1/e_2 w_2)\}$. Ultimately, this can be expressed as the amount of plants necessary for a predator to survive. If plant biomass $> II$ and $< III$, macroinvertebrates are the highest trophic level and TI =

2; if biomass $> III$ and $< IV$, small fish are the highest trophic level and TI = 3, and so on.

## 2.3 Environmental flow requirement

Although several physical factors affect the structure and characteristics of fluvial ecosystems, the majority of environmental flow evaluation methodologies focus on the single factor of river discharge and use the hydrological method for setting EFRs.

In the present study, we suggest a new conceptual framework for evaluating fluvial ecosystems that cannot be evaluated by hydrological parameters. It is based on the three characteristics, *productivity*, *ripple effect* and *resilience*. To evaluate these quantitatively, we proposed the respective indices: TI, CDE and ERT. Using these indices, we reclassified the criteria of an existing hydrological method and estimated EFR.

As a base model, we used the Tennant method, one of the most widely used hydrological methods for setting EFR

(Tennant, 1976). This method describes EFR as a percentage of mean annual discharge (MAD), an approach that we modified by using the three indices. EFR was calculated from a combination of base flow and disturbance using the following formulae:

$$\text{EFR} = x_1 + x_2 \tag{5}$$

$$x_1 = \left\{ x_{min} + \left( \frac{a+b}{2} \right) x_{range} \right\} \text{MAD} \tag{6}$$

$$x_2 = \alpha \text{MAD} \tag{7}$$





Here EFR is the environmental flow requirement ($m^3/s$), $x_1$ is the base flow ($m^3/s$), $x_2$ is the disturbance ($m^3/s$), MAD is the mean annual discharge ($m^3/s$), $x_{min}$ is the minimum flow requirement for MAD (%), $X_{range}$ is the range of flow for MAD (%), $a+b$ is a vulnerability index {a: rate of autonomy (0 to 1), b: ecological recovery time (0 to 1)}, and $\alpha$ is the disturbance coefficient. Since the Chikugo Model calculates NPP per annum, EFR in this paper represents the annual flow requirement. $x_1$ is calculated as the sum of $x_{min}$ and $x_{range}$ multiplied by the average of the vulnerability index. For example, if $x_1$ has a range from 30% to 60%, $x_{min}$ is 0.3 and $x_{range}$ is 0.3. By multiplying the vulnerability index (0–1) by $x_{range}$, $x_1$ varies from 30%, where priority is lowest, to 60%, where priority is highest. $x_1$ is set according to TI, while $x_2$ (disturbance magnitude) is set according to flow variability (maximum monthly flow/minimum monthly flow).

Criteria for $x_1$ are based on the Tennant method. Tennant (1976) suggests that the minimum flow for maintaining an aquatic habitat throughout the year should be no less than 10% of MAD, that >30% of MAD is necessary to maintain the habitat in fair condition, and that >60% is required for optimum conditions. For regions in which ecological structure is 'poor' (TI = 2), 10%–20% of MAD is required; 20%–30% for 'simple' (TI = 3) regions; 30%–60% for 'moderate' (TI = 4) regions; and 60%–100% for 'diverse' (TI = 5) regions. For 'uniform' regions (TI = 1), 3%–10% of MAD is set. Disturbance ($x_2$) is defined as a percentage of MAD, based on the criteria of the global model of Smakhtin *et al*. (2004). In that model, the 'high flow requirements' necessary for maintenance of the channel and natural morphology are given at 10%–30% of MAD, depending on the flow variability at different Q90 levels. In order to define several magnitudes of disturbance, we categorized catchments and regions into six hydro-climatic regions based on average monthly temperature and time series of specific discharges of 100 km$^2$ catchments. Extremely arid regions with specific discharges of < 0.03 $m^3/s$ were not considered here. Regions with flow variability (FV: maximum monthly discharge/minimum monthly discharge) above 1,000 were categorized as 'savanna'; FV 80–1,000 was categorized as 'monsoonal'; a specific maximum monthly discharge > 8 $m^3/s$ was categorized as 'wet moderate'; and maximum monthly discharge < 8 $m^3/s$ and average monthly temperature < 0 °C were categorized as 'snowmelt'. The rest of the world was categorized as 'moderate'. For savanna, characterized by strong flow variability, $x_2$ was set to 20% of MAD; for monsoonal regions it is 15% and the other regions where perennial flow is relatively steady, $x_2$ is set to zero.

## 2.4 Parameter setting

The model concept is shown in Figure 3. A grid setup is used to model the river channel network and adjacent terrestrial areas. Flow direction is indicated in each grid cell. In each cell, NPP is calculated from the provided climatic parameters. As NPP of aquatic vegetation is dependent on channel habitat size, channel habitat size was set in each cell using river width and length. River width is calculated from mean annual discharge, assuming that the water course had a triangular cross-section. River length is derived by multiplying grid cell length ($\Delta x$) and coefficient of meander ($\alpha$) (Figure 3 (a)), using the Chikugo model, a well-established approach to estimate annual NPP (Seino and Uchijima, 2010). NPP is calculated from net radiation, precipitation and annual average temperature. These climate data records from 2001 to 2010 was obtained from





WATCH Forcing Data methodology applied to ERA-Interim reanalysis data (WATCH, Weedon et al., 2011). Primary production is the main source of plant biomass, in addition to a certain amount (*C*) supplied from terrestrial vegetation. A portion of biomass enters the cell from upstream and leaves downstream. To simplify the model, we assumed that the amount of biomass flowing out of the cell corresponds to flow velocity and is defined it as the biomass amount passing along

the river length in a specific period. In addition to these processes, a portion of biomass disappears through decomposition and mineralisation (Figure 2 (b)). Theoretically, these processes occur simultaneously and the amount of biomass is maintained in equilibrium. Grazing by primary consumers (herbivores) was disregarded when estimating potential production.

The biomass accumulation process starts at the initial value of 0 and ends when change in B in every cell falls

below $1.0 \times 10^{-3}$ g. To test sensitivity to the initial value, with the model was run with several values from 0 to 1000. It was confirmed that the result was independent from the initial value. The fluvial biomass model was established on the basis of a global river channel network and catchment data from The 30' global drainage map (DDM30: Döll and Lehner, 2002) with a spatial resolution of $0.5° \times 0.5°$ ($360 \times 720$ grid cells worldwide). The monthly flow discharge from 2001 to 2010 was calculated by using H08 (Hanasaki et al., 2008). In reality, biomass accumulation processes are affected by climatic

conditions such as temperature, soil humidity and vegetation types; however, we applied single global parameters in order to demonstrate the basic characteristics of the model. It is known that the majority of plant biomass in the steep upper reaches of streams is supplied from terrestrial vegetation, while biomass sources are more strongly autochthonous in the wider middle to lower reaches (Begon, 1996, Odum, 2004). We implemented this general trend as a median value of *β = 0.5* for the proportion of allochthonous biomass across the entire watershed. The mineralization and decomposition parameter r was

set to 0.1 based on Sasaki (2011)'s study on the metabolism of submerged plants. For parameters of TI, we assumed the presence of target species common to Japanese rivers for each trophic level, and chose values with reference to existing studies on the metabolism of aquatic organisms (Table 3).

## 3 Results

### 3.1 Fluvial Biomass

Figure 4 illustrates biomass accumulation in rivers on a global scale. Our model estimates the total amount of fluvial biomass as $1.30 \times 10^8$ t, while previous estimates have ranged from $0.40 \times 10^8$ t to $14.5 \times 10^8$ t (Kira, 1976.; Whittacker and Likens, 1973). The results of our study, which estimates potential fluvial biomass without any human impacts, fall within this range. The calculated biomass is affected by the assumptions made about total global freshwater area, which was calculated by multiplying river width (calculated from mean annual discharge) by river length, which in turn is derived from multiplying

the length of a grid cell (*Δx*) by the coefficient of meander (*α*). In this study, *α* = 3 was used for all cells. Sensitivity checks were run with *α* = 2 and 4, yielding a biomass of $1.10 \times 10^8$ t in the former and $1.74 \times 10^8$ t in the latter case.





Fisher et al. (1973) reported that the annual amount of organic matter in a river channel increased by 1,600% if flooding occurred. Even if no noteworthy hydrological events occur, the amount of biomass may vary by ±20% in two successive years (Mulholland, 1981). Biomass in observed data have been reported as 0.06 to 1.17 t/ha in Warnow River in Germany (0.18 t/ha in this study) (Bahnwart, et al., 1998), 0.04 to 0.09 t/ha in Gamtooth Estuary in South Africa (0.14 t/ha) (Kotsedi, et al., 2012), 0.01 to 0.04 t/ha in Canadian Rivers (0.03 to 3.0 t/ha) (Bum and Pick, 1996), 0.21 to 0.76 t/ha in Tama River, Japan (0.29t/ha) (Aizaki, 1980), and 0.02t/ha in Streams of Amami Island, Japan (0.14t/ha) (Abe, et al., 2008). Also, with the maximum difference of +400% found for the Little Tennessee River, where an observed biomass of 1.56 t/ha at Coggin's Bend (McTammany et al., 2003) was estimated by the model as 0.35 t/ha. Considering the variable characteristics of fluvial biomass accumulation mentioned above, the fluvial biomass model provides values of acceptable accuracy and is suitable for the purpose of contrasting relative characteristics between regions. The greater the amount of biomass, the more consumer organisms it can support, and ultimately the more complex an ecosystem may be based on it. In order to evaluate this structure, the biomass calculated by the model represents a potential amount of primary production in the absence of grazing. It should thus be noted that in real ecosystems, the amount of biomass is maintained at lower levels through grazing and other interactions by consumers.

TI was calculated based on the fluvial biomass (Figure 5), ranging from 0 to 5. The lowest TI of 1 described 15% of freshwater surfaces and applied to the Arabian Peninsula, the Namib Desert, the Saharan periphery, eastern Siberia and central Asia. Areas of TI = 2 (1%) barely appeared at the edge of deserts and quickly carried over into regions with TI = 3 (14%) along deserts, in central Australia, north-eastern Eurasia and some regions at higher latitudes. About half (57%) of freshwater surfaces were categorized as TI = 4. The highest TI of 5 (12%) appeared along large river catchments and in large basins at low latitudes, covering the majority of the tropical rainforests in Amazonia, the Congo basin, the eastern part of the Great Dividing Range in Australia, and southeast Asia. The purpose of TI is not to precisely represent the actual configuration of the 'ecological pyramid' but to offer an index evaluating the amount of total biomass supported by primary productivity in a region. However, the calculated TI results showed good correlation with actual ecosystems in that in that high TI estimates of 4 or 5 were correlated with high primary productivity (Begon, 1996).

To check how much biomass is supplied from upstream, a TI considering only local primary productivity without longitudinal flux was calculated (i.e., $B_u$, $C$ and $F$ in Formula (1) are 0). When including longitudinal flux, TI increased at 5% of freshwater areas. TI increased by one level on the periphery of deserts in eastern Australia, Iran and (broadly) East and South Africa; it increased by two levels in the middle of the Darling River; by two to three levels in the Tigris, the Euphrates and in the Tarim Basin; and by four levels in the mid-reach of the Nile, in the Niger, on the northern shore of the Caspian Sea and in the lower Colorado River. In the lower Nile, TI increased from 0 to 3 when longitudinal transportation of biomass was considered. This increase in TI was most obvious in rivers in arid to semi-arid regions, and it demonstrates that biomass supply from upstream and terrestrial vegetation makes it possible for regions with poor primary productivity to attain higher trophic levels than would be possible through local productivity alone. The model thus successfully expressed the effects of a river continuum that transports ecological richness beyond climatic barriers as a basis of primary productivity. In deserts



and arid region, where TI is 0–1, ecological richness is poor and annual discharge is low. Thus, EFR assessment may not be necessary for these regions. In contrast, in savannas, Asian monsoonal regions and along large rivers, where TI is 4 or 5, a high ecological richness is present; however, in most of these regions environmental flow is not considered in water resource management. EFR assessment should be allocated a higher priority in these localities.

## 3.2 Contribution to Downstream Ecosystems

CDE was evaluated by calculating the amount of biomass supplied downstream by a specific river section (Figure 6). CDE tended to be higher along large rivers where several tributaries merge and rivers in lower latitudes where NPP is high. The former example included the mid- to downstream Mississippi, Mackenzie, Amur, Volga, Yenisei and Lena, and major rivers in Western Europe. The latter included the upper Nile, Congo and Niger, the entire catchment of the Amazon, the Parana, and major rivers in India, eastern Asia and southeast Asia. In large rivers in high latitudes where NPP is lower, the confluence of tributaries and lakes plays an essential role in supporting downstream ecosystems, while in low-latitude rivers, high NPP can sustain relatively autonomic ecosystems. CDE also varied within catchments, and 'hotspots' serving as a biomass source appeared at the confluence of tributaries, lakes and areas with higher NPP. Although such spots may occupy only a small part of the entire catchment, they play an important role in supporting downstream ecosystems. Regions with higher CDE should be accorded high priority for ecological protection and consequently be allocated a higher EFR.

## 3.3 Ecological recovery time

Estimates ERT on a global scale, estimated at a range of 1–300, are shown in Figure 7. ERTs below 20 applied to 10% of total freshwater areas, estimates of 10–50 to 22%, of 50–100 to 53%, and of more than 100 to 14%. Areas of short ERT (<10) appeared in the uppermost reaches of streams in high mountains such as the Rocky Mountains, Alps and Himalayas, on tropical islands with high NPP such as central Papua New Guinea, Indonesia, and eastern Madagascar, and in the lower reaches of small rivers (channel length < 30 km) in temperate climates. ERT tended to be shorter where CDE was high (>0.8). The latter regions are characterized by highly autonomic ecosystems where biomass produced by the local NPP accumulates and is decomposed *in situ*. This allows the ecosystem to recover quickly without biomass input supplied from upstream.

Areas of long ERT (> 100) appeared along the mid- to lower reaches of large rivers such as the Mississippi, Amazon, Parana, Nile, Niger, Ob, Yenisei, Murray-Darling, Ganges and Yangtze. CDE tended to be higher (indicating autonomic conditions) in the lower reaches of large rivers (Figure 6); however, ERT concurrently was relatively longer. This may be because only a small proportion of allochthonous biomass is transported downstream across long distances, while most biomass originates from local NPP.





We compared the characteristics of ERT in the Mekong, for three cases: point destruction (case 1) and partial (up to mid-reach, case 2) and complete catchment scale destruction (case 3). Figure 8 shows the results. In case 1, 60% of biomass recovered within one year, and the system returned to a balanced state after 6 years. Here, biomass of non-impacted upstream areas quickly resupplied the cell. In case 2, the recovery process was gradual. It took more than 20 years to recover

90% of the original amount of biomass and 30 years to return to a balanced state. Case 3 followed a similar pattern to case 2, displaying the same recovery rate until year 27. Although the difference from case 2 is very minuscule and not actually visible in the illustration, after the year 27 the recovery rate of case 3 became slightly lower, ultimately requiring 50 years to recover the initial state. More than 90% of biomass contributed to the target cell (near Phnom Pen) originated below the midstream region, and consequently, destruction of this area had a great impact. In addition, 3% of biomass was supplied

over time from the upper reach.

The results indicate that ERT is affected by NPP, length of channel, and location/extent of disturbed area. Higher priority should be placed on EFR for regions where ERT is longer. In actual ecosystems, recovery of riverine vegetation after destruction by floods or dredging tends to be faster in a humid climate and slower in arid regions (Sloan et al., 2001). With the exception of the region at high latitudes, alpine regions and arid areas where NPP is extremely small, larger NPP

resulted in shorter ERT in this model, because NPP was calculated from annual precipitation. This is realistic in that actual recovery processes are affected by humidity. A recovery time of 20 years has been reported for instream vegetation in northern California, an exceptionally long time for a humid-temperate climate (Sloan et al., 2001). ERT in our model ranged from 20 to 40 years for the region in question, corresponding to actual observations. The model can thus be regarded as offering reasonable indicators for evaluating the resilience of fluvial ecosystems.

### 3.4 Environmental Flow Requirement

The global EFR map is shown in Figure 9. Average EFR of global freshwater areas was 42% of MAD. EFR of 0–20% appeared in only a small part of desert peripheries and arctic regions. EFR of 30% appeared in Siberia, Scandinavia, Canada, alpine regions and arid regions such as central Australia, central Asia and the upper Yellow River. EFR of 40%−50%

appeared in California, southwest Africa, the Middle East in the vicinity of Iran and the Caspian Sea, the Tibetan plateau, and large areas of western Eurasia. In addition, large rivers in semi-arid and monsoonal regions, such as the Nile, Orange, and Euphrates fell within this range. Higher EFR (>60%) emerged in tropical and sub-tropical regions at lower latitudes and along large rivers such as the Amazon, Parana, Mississippi, Tigris, Yenisei and Lena, Rhine and Elbe, Yangtze, Murray-Darling, and Ganges. The highest EFR (>80%) was particular to the lower reaches of the Mississippi, Amazon, Niger,

Ganges, Mekong and Rhine. These regional differences are attributed to magnitudes of disturbance. Average TI was 4.3 in Southeast Asia and South America. In these regions, a high ecological richness is correlated to a wide variability in seasonal flow. EFR is higher in monsoonal and savanna regions than tropical wet regions (central Africa, Indonesia and New Guinea) because disturbance ($x_2$) is higher for these regions. On the other hand, EFR is also high in large rivers outside monsoonal



and savanna regions. These regions are characterized as having low resilience, resulting in longer ERT; to protect these vulnerable ecosystems, a higher EFR was allocated. EFRs in previous study have been reported as 15%–20% in Sweden (40% in this study) (Renöfält et al., 2009), 25% in Canada (27%) (Linnansaari et al., 2012), and 19% in India (74%) (Joshi et al., 2014) and 10% in Turkey (51%) (Karakoyun et al., 2016).

## 4 Discussion

To put the outcomes of this study into context with the current state of research, a previous study (Smakhtin et al., 2004) was reviewed and compared with our results. Smakhtin presented a pilot global model for EFRs. This model was the first attempt to set global EFRs and has been widely applied in water resource assessments. The great advantage of this model is that it

offers global perspectives on environmental flow without depending on field observation and setting complex parameters. There are however some limitations to this model. First, it derives EFR from hydrological parameters only: primarily MAD, which defines low-flow requirement (LFR), and Q90 (the flow level exceeded 90% of the time), which defines high-flow requirement (HFR). However, the model cannot account for the possibility that ecological richness and vulnerability may vary depending on climatic characteristics and primary productivity even when MAD remains constant. Second, because the

majority of rivers have been degraded to some degree by human activities, the model seeks to compensate by setting a lower conservation threshold, deemed a more feasible goal for compromising with human demands. This concept may be inadequate for protecting riverine ecosystems. Third, while climatic conditions and characteristics of tributaries may affect biological diversity within a catchment, the model is unable to distinguish these differences at the catchment scale.

To highlight the characteristics of our model, we calculated the difference in estimated EFR between our model and

20 Smakhtin's model (referred to in the following as the pilot model) (Figure 10). EFR in the present study was on average 14% higher than in the pilot model. In particularly, EFRs were noticeably higher in savanna and monsoonal regions. This tendency may be explained by the fact that our method set EFRs relative to potential ecological richness in the absence of any human impacts, while the EFR of the pilot model set more feasible goals for degraded rivers. Several other possible explanations may be noted. The pilot model hypothesized that in regions with a wide range of seasonal flow variation,

fluvial ecosystems are resistant to long droughts, and premising from this that the ecosystems would also be resistant to anthropogenic impacts to some degree. In contrast, where discharge is stable and abundant, a decrease in discharge through flow regulation may have a larger impact on ecosystems, resulting in larger EFR being allocated for the region (Smakhtin et al., 2004). Based on this hypothesis, the pilot model applied high EFR (> 40%) to tropic rivers, western Europe and the higher latitudes of North America, where the annual flow regime is relatively stable and abundant. On the other hand, EFRs

were relatively low (15%–30%) in eastern to south-eastern Asian monsoonal regions, savanna regions in Africa and Australia, and some parts of South America where there exists high variability in annual flow. In contrast to the pilot model, our model found that there were regions where ecological richness was relatively poor although annual discharge was stable



and abundant. At the same time, it was found that there were regions with high ecological richness where the ecosystem was more vulnerable because of high flow variability. These regions should be considered as having high priority protection requirements, and higher EFR should thus be allocated. They consist of Asian monsoonal and savanna regions and the Amazon basin, the latter including high-latitude lakes and the confluences of large rivers. This result was in notable contrast

to the hypothesis of the pilot model. Our findings suggest that this type of regions should receive 20%−50% higher EFR than has previously been proposed.

Figure 11 illustrates details of the EFR assessment for the Nile and the Yellow River as calculated in our model. In the Nile, EFR was ~89% in the upper reach and became 68% at the confluence with the Sobat at Malakal, subsequently dropping to 35% at Aswan and 33% at Cairo. In the Yellow River, in contrast, EFR ranged from 35% to 55% and remained

relatively even up to the confluence with the Wei at Dongguan, after which it increased to ~90%. Both rivers have a wide range of EFRs throughout their length, which may be explained by the large variations in TI, DS and ERT. The upper Nile is situated in savanna with high productivity and the potential for downstream ripple effect (TI = 5, CDE > 0.8), leading to allocation of a high EFR. After the confluence with the Blue Nile, as the river enters a low-production arid region and lacks confluences with large tributaries (TI = 3, CDE < 0.1), EFR was lower. In the Yellow River, the upper reach flows through a

low-production arid region, while the lower reach is located in a highly productive wet-moderate climate zone.

In addition, tributaries originated from three different climatic regions (snowmelt, moderate and wet-moderate) with different disturbance values, and tributaries with different resilience and ripple effect values affect the CDE and ERT of the main stream. In the pilot model, only one EFR was set for each river basin (24% for the Nile and 31% for the Yellow River). Overall, EFR in the pilot model was at approximately the same level as the minimum in our model (30%). This may imply

that the pilot model was able to evaluate rivers with simple ecosystems, but possibly underestimates for regions with high ecological richness and vulnerability. The advantage of our model is that it is able to set different EFRs within a river basin based on differences in ecological characteristics, driven by climatic conditions and patterns of tributaries.

Expressed another way, EFR can be seen as representing the amount of water that can be taken from a river. When EFR decreases in the lower reach, as in the Nile, water available for extraction will be calculated automatically. When EFR

increases in the lower reach, as in the Yellow River, it is necessary that the upper reach also preserve EFR for the lower reach. Consideration of such longitudinal differences in EFR aid in setting integrated water allocation strategies by reflecting differences in water resource availability for humans within a catchment.

## 5 Conclusions

*Productivity*, *ripple effect* and *resilienc*e are essential characteristics of fluvial ecosystems. In order to evaluate these three

factors and apply them as criteria in EFR evaluation, we propose a fluvial biomass model which calculates the amount of biomass (productivity) accumulated through physical and climatic processes. Using this model, we introduce the indices Contribution to Downstream Ecosystems (CDE) and Ecological Recovery Time (ERT) to evaluate ripple effect and





resilience, respectively. We then improve the Tennant EFR method based on these ecological characteristics and estimate

EFR on a global scale. In comparison with existing global estimates, which were based on flow regime only, the present
model suggests 20%–50% higher EFR in monsoonal and savanna regions with high ecological richness, and for the lower
reaches of large rivers in higher latitudes where primary productivity is low and the ecosystems largely depends on
allochthonous biomass supply (i.e., where the ecosystem is less resilient). The main advantage of our model is the ability to
set different EFRs within a river basin based on differences in ecological characteristics driven by climatic conditions and
tributaries. Taking such longitudinal differences in EFR into account enables the development of integrated water allocation
strategies by reflecting differences in water resource availability for humans within a catchment. Since it is based on a simple
and well-understood mechanistic fluvial ecosystem model, the approach may be adapted to different regions by adjusting
parameters based on local conditions. We hope that this approach will provide a useful perspective in determining
appropriate strategies for sustainable water resource management. Planned further developments of this model include
expressing EFR on a monthly basis and considering fluctuations of primary productivity according to disturbances and
multiple climatic and biophysical influences. Regional parameters also need to be tested and checked for validity. These
issues will be treated in a future study.

**Data availability**

The data used in this study are freely available from different sources. Climate data records from 2001 to 2010 (temperature,
precipitation, and net radiation) are available online from EU WATCH (Water and Global Change: http://www.eu-
watch.org/watermip/data-format). The river discharge was simulated by open source of global hydrological model with
human activities (H08: http://h08.nies.go.jp/h08/index.html). The 30' global drainage map (DDM 30) are free to download
from Goethe-Universität Frankfurt am Main (https://www.uni-frankfurt.de/45217896/3_drainage_direction_map).

**Acknowledgements**

We would like to thank Dr. Hanasaki, N, National Institute for Environmental Studies, Tsukuba, Japan for offering the H08
data set and helpful advice for developing the model.

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





**Figures**

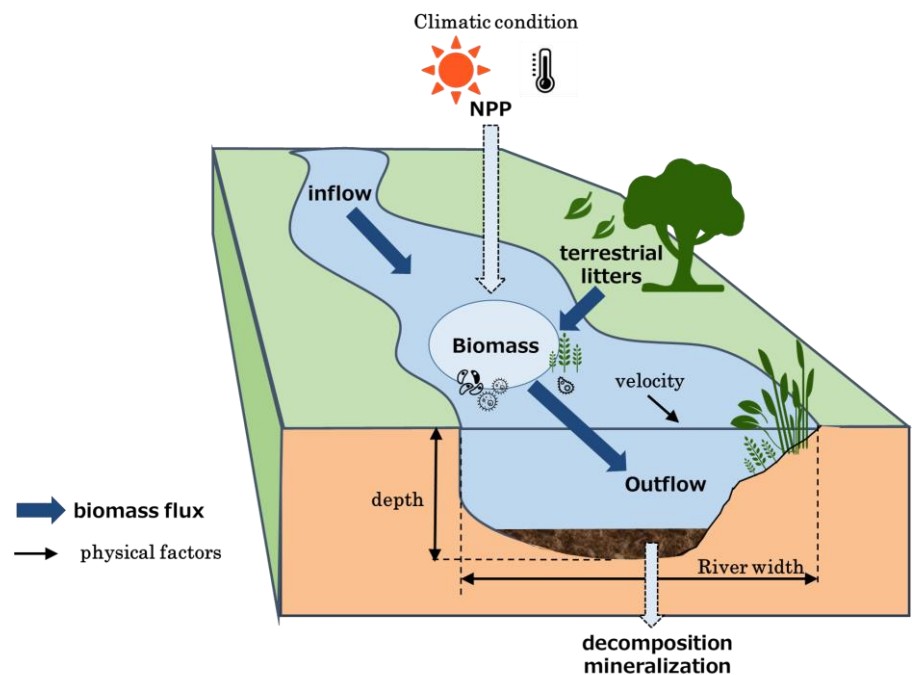

**Figure 1.** Conceptual representation of the fluvial biomass model

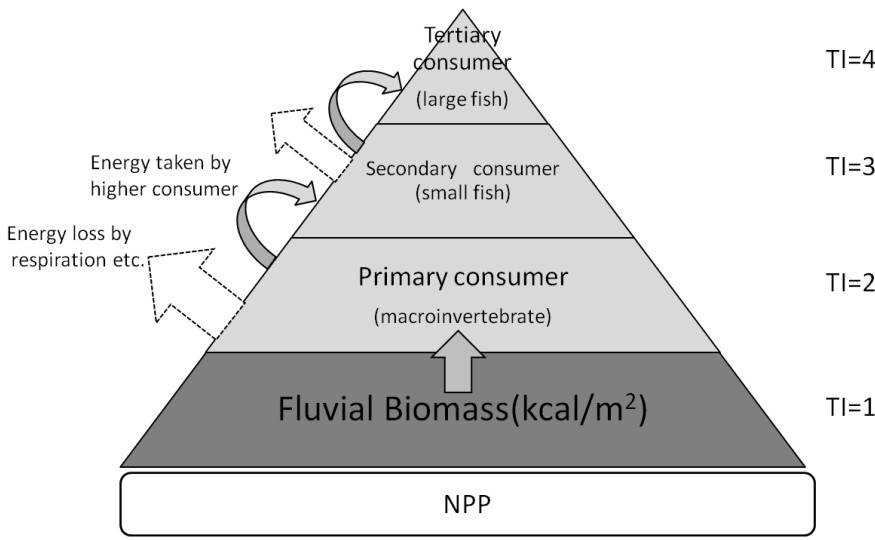

**Figure 2.** Conceptual basis of trophic level index (TI)





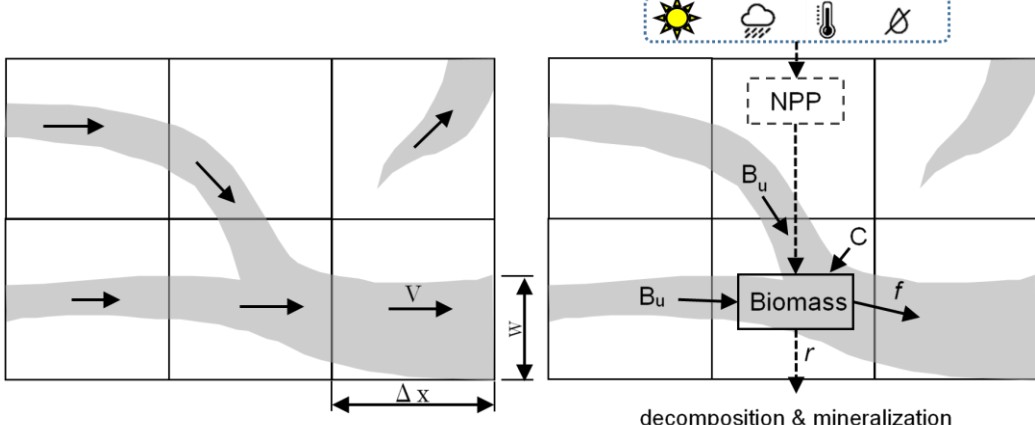

**Figure 3.** Grid layout used in the model. (a) Grid parameters. $\Delta x$: cell length; $w$: stream width; $V$: flow velocity. (b) Inputs and outputs of a grid cell. $B_u$: allochthonous upstream biomass; $C$: allochthonous terrestrial biomass; $f$: flux rate; $r$: decomposition/mineralization rate.

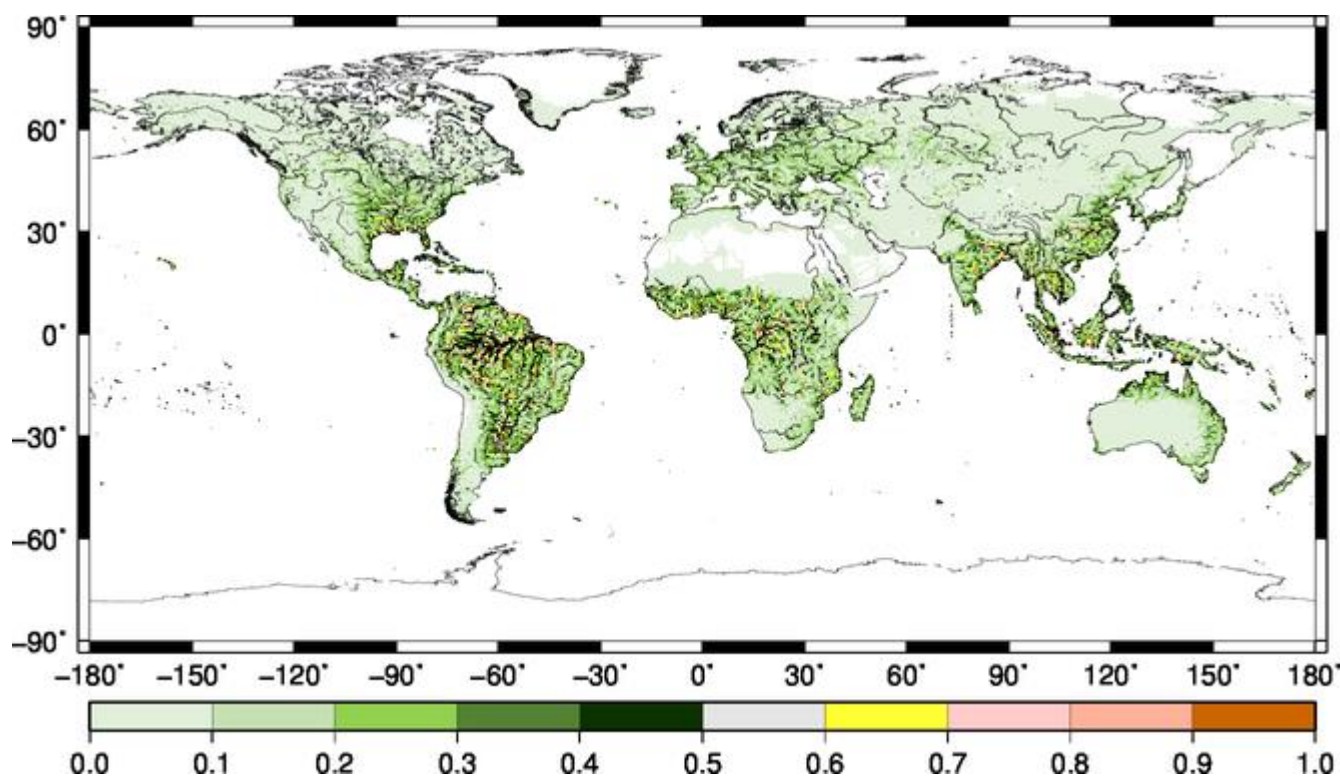

**Figure 4.** Calculated global fluvial biomass(t/ha)



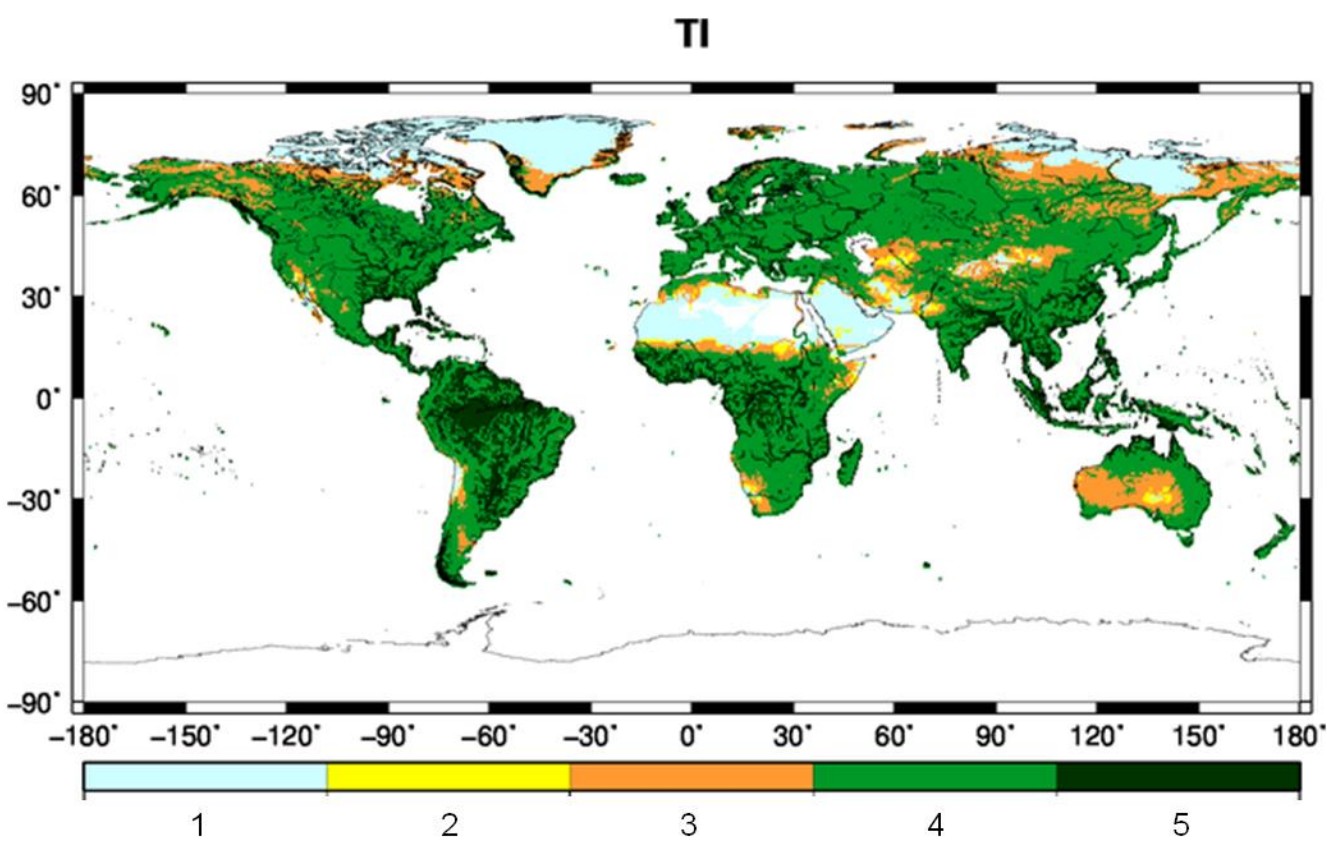

**Figure 5.** TI estimates on a global scale (the illustrate shows with longitudinal flux)





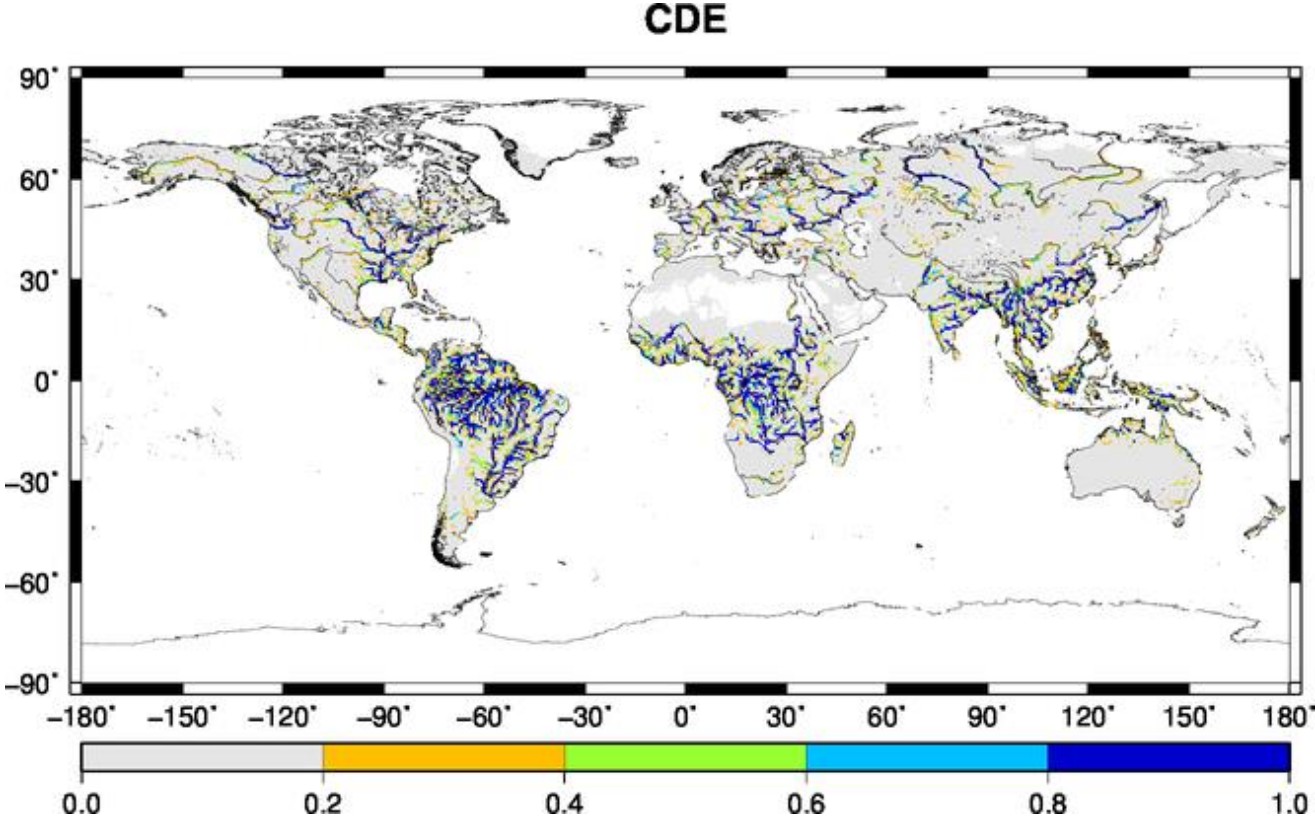

**Figure 6.** Contribution to downstream ecosystems (CDE) estimates on a global scale



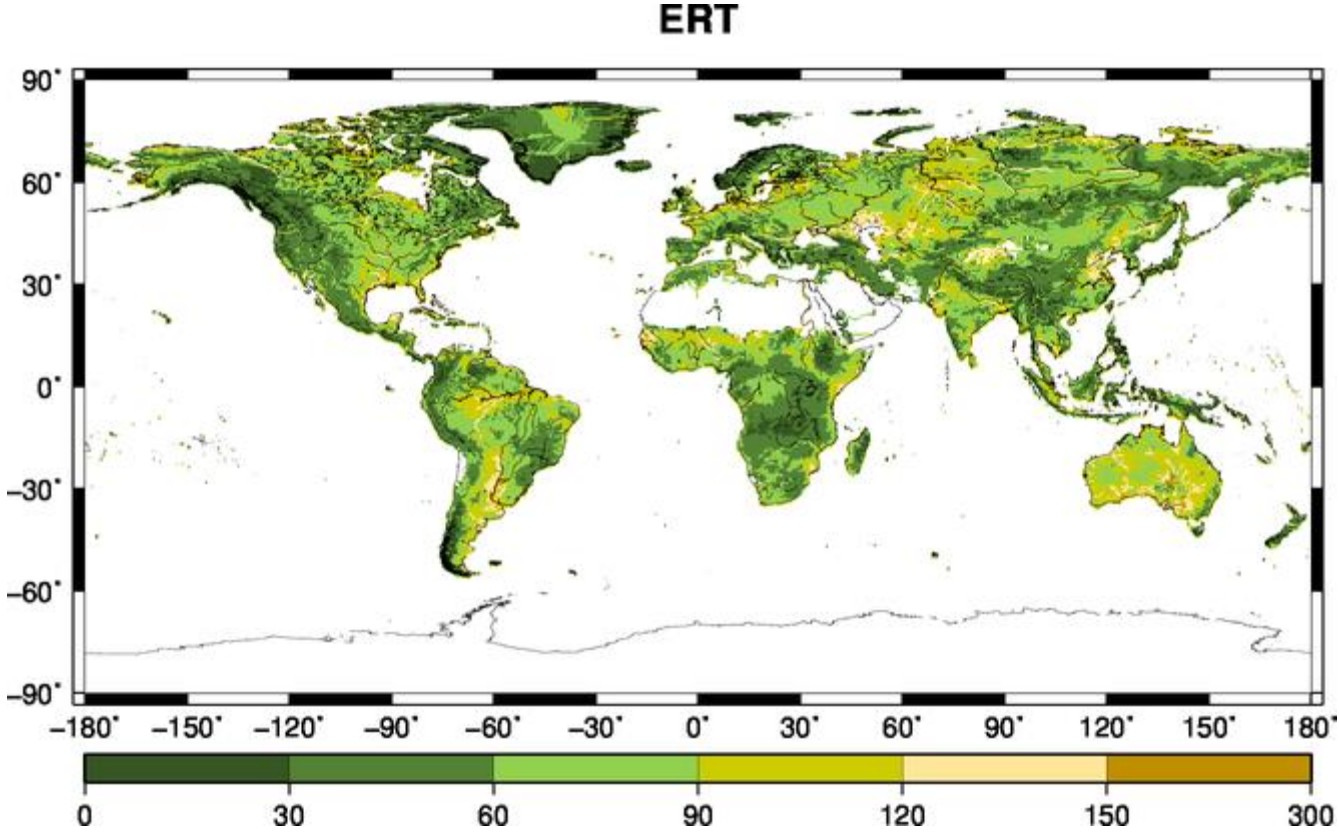

**Figure 7.** Ecological recovery time (ERT) estimates on a global scale

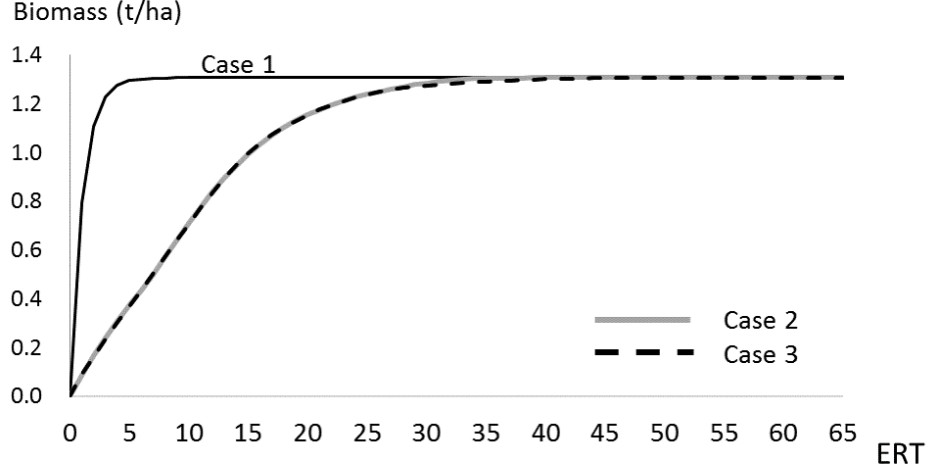



**Figure 8.** ERT for different scales of vegetation destruction (lower Mekong at Phnom Penh). Case 1: point destruction, case 2: partial catchment destruction, case 3: whole catchment destruction.

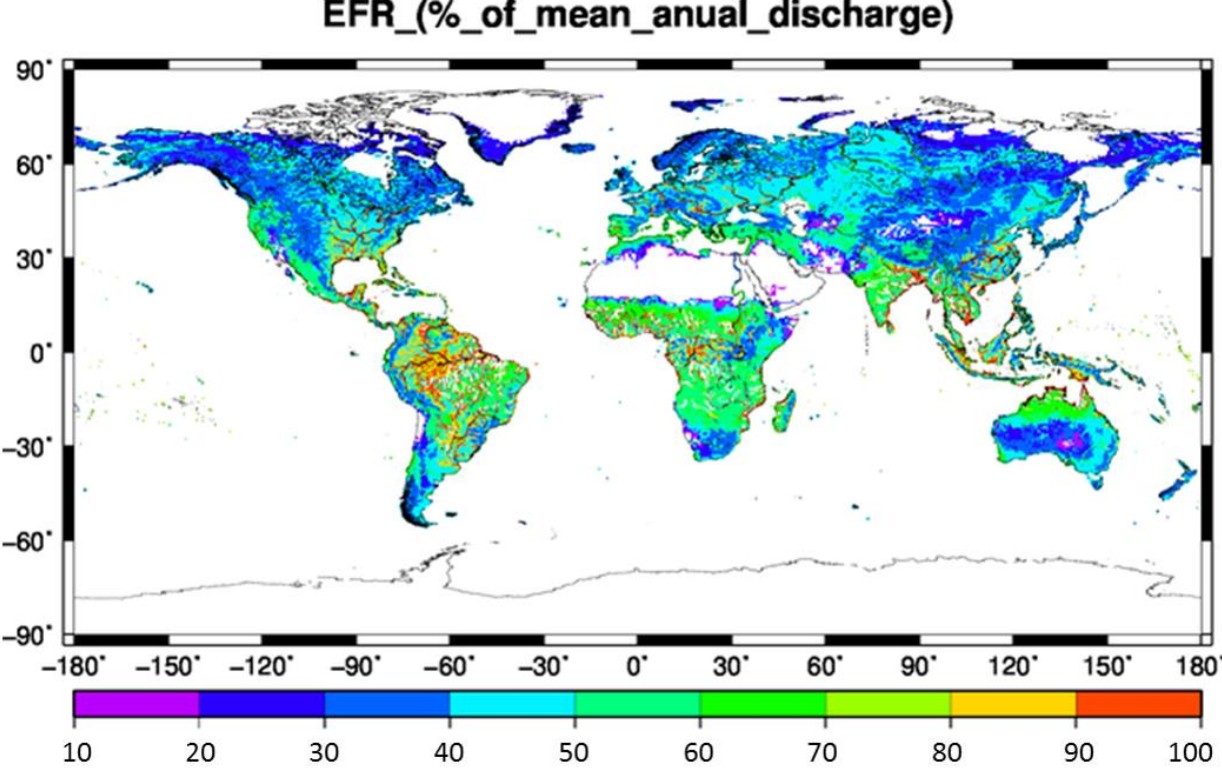

**Figure 9.** Environmental flow requirement (EFR) estimated on a global scale





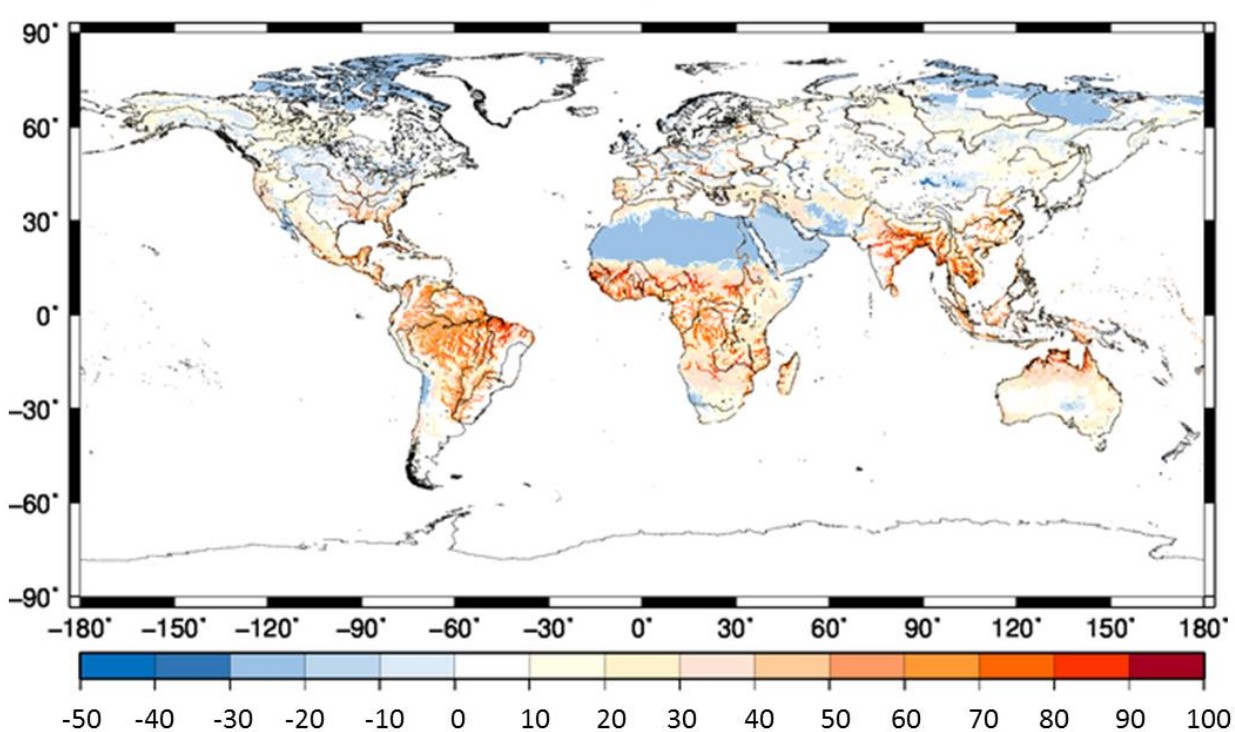

**Figure 10.** Difference in EFR between the present model and Smakhtin et al. (2004) on a global scale

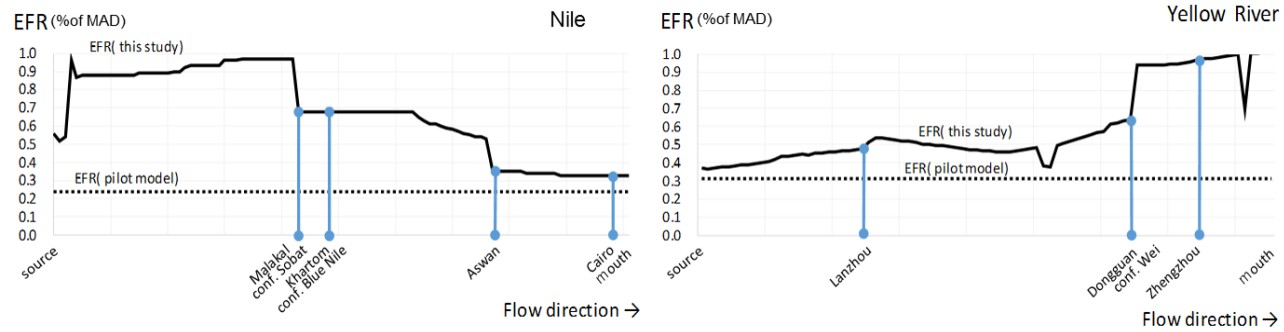

**Figure 11.** Longitudinal changes in EFR in the main channel of the Nile and Yellow River. The river length from source to mouth is 6,853 km (Nile) and 5,464km (Yellow River) respectively.



## Tables

**Table 1.** Parameters to evaluate trophic level index (TI)

| Trophic level index (TI) | 0 | 1 | 2 | 3 | 4 | 5 |
|---|---|---|---|---|---|---|
| Target species | none | plants | macroinvertebrate | small fish | mid-sized fish | large fish/bird |
| Basic metabolic rate of target species(kcal/g/day) | - | - | $\varphi_2$ | $\varphi_3$ | $\varphi_4$ | $\varphi_5$ |
| Energy of target species(kcal/g) | - | $e_1$ | $e_2$ | $e_3$ | $e_4$ | $e_5$ |
| Efficiency of energy intake (%) | - | - | $\delta_2$ | $\delta_3$ | $\delta_4$ | $\delta_5$ |
| Weight of target species (g/individual) | - | - | $w_2$ | $w_3$ | $w_4$ | $w_5$ |

5  **Table 2.** Calculation processes for estimating TI

| Trophic level index (TI) | 2 | 3 | 4 |
|---|---|---|---|
| Target predator | macroinvertebrate | small fish | large fish |
| Energy necessary for an individual to survive (kcal/individual/year) | $\dfrac{365\varphi_2 w_2}{\delta_2}$ | $\dfrac{365\varphi_3 w_3}{\delta_3}$ | $\dfrac{365\varphi_4 w_4}{\delta_4}$ |
| Energy of an individual transferred to a predator of higher trophic level (kcal/individual) | $e_2 w_2$ | $e_3 w_3$ | $e_4 w_4$ |
| Number of individuals of lower trophic levels necessary for the predator to survive (individual) | $-^{1)}$ | $\dfrac{365\varphi_3 w_3}{\delta_3} \times \dfrac{1}{e_2 w_2}$ | $\dfrac{365\varphi_4 w_4}{\delta_4} \times \dfrac{1}{e_3 w_3}$ |
| Minimum amount of plants necessary for predator to survive (g/individual) | $II$ $= \dfrac{365\varphi_2 w_2}{\delta_2} \times \dfrac{1}{e_1}$ | $III$ $= \left(\dfrac{365\varphi_2 w_2}{\delta_2} \times \dfrac{1}{e_1}\right)$ $\times \left(\dfrac{365\varphi_3 w_3}{\delta_3} \times \dfrac{1}{e_2 w_2}\right)$ | $IV$ $= \left(\dfrac{365\varphi_2 w_2}{\delta_2} \times \dfrac{1}{e_1}\right)$ $\times \left(\dfrac{365\varphi_3 w_3}{\delta_3} \times \dfrac{1}{e_2 w_2}\right)$ $\times \left(\dfrac{365\varphi_4 w_4}{\delta_4} \times \dfrac{1}{e_3 w_3}\right)$ |
| TI | $II \le B^{2)}$ , TI=2 | $III \le B$ , TI=3 | $IV \le B$ , TI=4 |

1) Plants are not considered as individuals but as vegetated area (m²).
2) B: plant biomass (g/m²)





**Table 3.** Parameters for calculating TI

| TI | 1 | 2 | 3 | 4 | 5 |
|---|---|---|---|---|---|
| Target Species[1] | Plants/Algae | Macroinvertebrate (*Chironomidae*) | Small fish (*Oryzias latipes*) | Middle fish (*Tribolodon hakonensis*) | Large fish (*Opsariichthys uncirostris*) |
| $\varphi$(kcal/g/d)[2] | - | 0.051 | 2.000 | 2.000 | 2.000 |
| $e$ (kcal/g) | 4.2[3] | 5.0[4] | 3.0[5] | 3.0 | 3.0[6] |
| $\delta$(%)[7] | - | 0.1 | 0.1 | 0.1 | 0.1 |
| $w$ (g/individual) | - | 0.01 | 1.00 | 50.00 | 500.00 |

1) Common Japanese species were assumed as representative for each trophic level. 2) Basic metabolic rate [W (kcal/day)] was calculated using the following formula (Peters, 1983), which is generally applicable to poikilotherms: W = 20.7×0.14× (body weight [kg]) ^0.751, 3) Average energy of 57 plant species (Golley, 1961). 4) Average energy of insects in general (Mihashi, 2012). 5) MEXT, 2016, 6) As no data for *Opsariichthys uncirostris* was available, the value for a *Cyprinus carpio* of similar body size and belong to the same family was used (MEXT, 2016). 7) Odum, 2004.