# Peer review of "Evaluating primary productivity, ripple effect and resilience of fluvial ecosystems: a new approach to assessing environmental flow requirement"

_Hydrology and Earth System Sciences, 2016_

## Referee Comment (RC1) · M. McClain (Referee) · 29 Jan 2017

This paper presents a new model for the estimation of environmental flow requirements (EFR) globally using a simulated raster river channel network with a spatial resolution of 0.5x0.5 degrees. The model differs from past models by incorporating biomass and biomass fluxes in setting the environmental flow requirement. Biomass increases in any cell via modelled inputs from a global NPP model, calculated inputs from adjacent terrestrial areas, and a calculated input from the upstream cell. Biomass decreases in a cell via decomposition and transport to the downstream cell. The influence of

biomass in EFR calculations is expressed by three indices, the Trophic Index (TI), contribution to downstream ecosystems (CDE), and ecological recovery time (ERT). The TI reflects the number of trophic levels present (1 to 5), which is expressed as ecological structure ranging from 'poor' (TI=2) to 'diverse' (TI=5). These levels are then interpreted to represent the levels of general condition of streams presented in the method of Tennant (1976) ranging from 'poor' to 'optimum' condition. TI is determined using a tropic model estimating the number of potential trophic levels based on the amount of biomass calculated in the fluvial portion of the model cell. This index is used to set an initial minimum and range of low flows that are subsequently modified based on the values of the remaining two indices. CDE is based on flow velocity and normalised such that CDE approaches zero for the slowest modelled velocities and 1 for the highest modelled velocities. ERT is calculated as the time (model annual time steps) required to reach stable biomass equilibrium if biomass is reset to zero. ERT is also expressed in the EFR calculation as a normalised value between 0 and 1, with 1 reflecting the fastest recovery time. The combined effect of CDE and ERT is that, as they increase in value, low flow requirements are set progressively closer to the maximum value of the low flow range, as originally set by Tennant (1976). A high flow component of the ERT is also calculated based on the annual variability in monthly mean flows, similar to the approach used previously. Higher intra-annual variability results in larger values for the high flow component.

The authors refer to their model as an improvement of the Tennant method. They also state that "Instead of offering specific values and criteria, in this paper, we provide a perspective on the conceptual method for setting EFRs." Specific values are, however, eventually offered and compared with previous results of Smakhtin et al. (2004), and the outcome of the comparison is presented in the conclusions and abstract.

The authors are to be commended for their efforts to add new, ecologically relevant variables (TI, CDE, and ERT) to the quantification of EFRs at a global scale. I do, however, believe there are significant flaws in the current formulation of the approach

which limit the applicability of the model and its results.

My main concern is that the authors have misconstrued the meaning of the incremental levels of flow identified in the Tennant method. This is critical because the selection of the low flow minimum and range (x1 in their calculation) exerts a dominant control on the eventual EFR. CDE and ERT simply adjust low flow recommendations within the range established by Tennant. The authors have interpreted that the incremental levels reflect variable ecological structure (i.e. increasing number of trophic levels), when in fact the levels reflect the condition (from natural to increasingly degraded) of the river ecological structure and function, whatever the natural state might be. This follows from a simplified view of the natural flow paradigm that the risk of degraded ecological condition increases as anthropic flow alteration increases. Instead, the authors assume that the incremental levels relate to the natural levels of ecological structure. The implication is that, in the author's approach, systems which have increasingly simpler ecological structures are expected to maintain equivalent ecological condition (relative to natural) with increasing anthropic flow alteration. I know of no ecological theory or research to support this assumption. If the authors are to continue with this approach they should present a clear theoretical justification and supporting research results.

Their assumption also erases from the calculation of ERF the essential (societal based) process of setting objectives for ecosystem management, such as the requirement of achieving 'good' ecological status in all water bodies of the European Union. The levels set by Smakhtin et al. (2004) considered environmental management objectives (as required by best practice in setting ERTs), but the present paper does not. I recommend that the authors take note of this omission of management objectives in their approach and consider ways to rectify it.

In addition to these concerns, I offer the following comments for the authors to consider in the revision of their manuscript.

Introduction:

Pg. 2, line 13: parts of the text beginning "Early approaches aimed to define. . ." appear to be copied and pasted or slightly modified from Pahl-Wostl et al. 2013. Any copied and pasted or slightly modified text in the manuscript should be deleted. Citing the source of copied and pasted text is not sufficient. All text not contained in quotation marks must be original and attributable to the authors alone.

Pg. 3, line 9: "Stream flow has often been treated as the 'master variable' since it can be readily described by indices" This is not the reason stream flow is considered as the master variable. Revert to the original source (Power, Mary E., et al. "Hydraulic food-chain models." BioScience 45.3 (1995): 159-167) to clarify.

Pg. 4, line 7: Tharme 2003 is not correct reference for IFIM. Check and correct alignment of methods and original sources throughout paper.

Method:

Pg. 4, line 25: this section begins with the repetition of points made above. In fact there is quite a bit of redundancy throughout the manuscript that should be removed.

Pg. 5, lines 3 and 4: change PRC to RPC.

Pg. 9, TI section: the calculation of TI using this approach is overly detailed for the global scale and approach of the model. I recommend seeking a much simplified approach, taking into consideration my main concerns expressed at the beginning of this review.

Pg. 10, beginning line 9: as mentioned in the initial paragraphs of this review, the authors have misconstrued the purpose of Tennant's incremental levels of river ecological condition. Please review Tennant's paper carefully and represent accurately in this paper.

Pg. 10, line 20: the switch from ratios (FV 80-8000) to flow magnitudes (>8m3/s) is unexpected and unexplained here. Is it correct?
Pg. 10, line 32: first mention of the Chikugo model for quantification of NPP. This is at the root of all following calculations but is not well described. First, the indication throughout the paper is that fluvial NPP is being calculated, but from what I read in Seino and Uchijima 1993 (not 2010), the model calculates terrestrial (or generic) NPP. How is 'fluvial' NPP calculated? If the model is not 'fluvial' specific there should be an explanation of the rationale the authors use for the model. Also, the model is described as "well-established" (line 32) but according to Google Scholar Seino and Uchijima (1993) has been cited only 5 times in 26 years. What is the rationale for "well-established"?

Results:

Pg. 11, line 30: $\alpha$ is set as 3 globally, and length of grid cell is also the same, Therefore, it seems length is removed as a variable globally. Are there consequences to this simplification?

Pg. 13, line 12: tributaries and lakes are indicated as significant in influencing the results, but I do not understand how these are resolved (made significant) in the model. Are these resolved somehow independently in the 0.5x0.5 degree grid? If so explain.

Pg. 13, line 30: use of language like "this may be because. . ." suggests that the processes and relationships controlling ERT are not fully understood, but they are exactly known as represented in the mathematics of the model. Refer to the relationships in the model and explain more confidently.

Pg. 15, line 1: "These regions are characterised as having low resilience, resulting in longer ERT. . ." What is the research evidence (papers?) for the lower resilience of large rivers outside of monsoonal regions and savanna regions? The authors have defined resilience based on ERT. Do not turn this around and assume low resilience because of the ERT calculated.

Discussion:

Pg. 15, line 9: provide citations supporting that Smakhtin et al, results have been "widely applied in water resource assessments".

Pg. 15, line 23: Reference is made here to "feasible goals", which refer back to my concern about the lack of environmental management objectives in the approach of the authors. This needs more explicit attention in future versions of the model.

Conclusions:

Pg. 17, line 1: "We then improve the Tennant EFR. . ." as mentioned above I believe the authors have misconstrued the incremental levels of condition in Tennant, therefore I believe the Tennant method has been misused and not improved. Substantial attention is needed to address this in future versions of the model.

---

## Referee Comment (RC2) · S. Wenger (Referee) · 30 Jan 2017

This ambitious manuscript attempts to develop a global model for environmental flow requirements built on a key ecosystem function: primary production. This is a very interesting idea, and I appreciate the authors' efforts, but I do not find the proposed model at all convincing. The authors have almost completely ignored the relevant literature on ecology in general and river metabolism in particular that should serve as a foundation of this effort; in fact, they never even mention ecosystem metabolism in the manuscript. They rely on a few classic studies to set the stage, but seem unaware

of the many empirical and theoretical contributions that have come in recent decades (though some studies are mentioned in passing). Consequently, they coin new terms for concepts that already exist, and propose indices and relationships in an almost complete scientific vacuum. The section on the "trophic level index", for example, should be built on the rich existing literature on trophic ecology, metabolic theory, food chain length, energetics, etc; it cites only an Ecology textbook.

I don't want to discount the hard work of the authors, and there's sometimes a role for blue-sky thinking unencumbered by others' ideas, but this is not the way to move science forward. I think the authors should consider convening a group of colleagues with diverse knowledge in the relevant scientific fields to develop an interdisciplinary solution to the problem informed by the relevant literature.

---

## Short Comment (SC2) · 13 Feb 2017

The authors brought new insights into the global eco-hydrology field by considering NPP, resilience and trophic levels. These last ideas are necessary to improve eco-hydrology relationships at global scale. However, these last deserve more investigation, monitoring and testing before being upscaled to global EF methods.

I recommend to test their method with empirical case study first and second to acknowledge worked developed by (Gerten, Hoff et al. 2013, Pastor, Ludwig et al. 2014) with the development of which the VMF method which was not mentioned in this review, neither the Q90_Q50 and Tessman methods which shown better results than the Smakthin and Hanasaki and Tennant methods (Pastor, Ludwig et al. 2014). The VMF method was acknowledged by other global assessments including: (Gerten, Hoff et al. 2013, Boulay, Bare et al. 2015, Gaupp, Hall et al. 2015, Sadoff 2015, Steffen, Richardson et al. 2015) and was used to defined freshwater planetary boundaries in science (see references). This study is based on the Tennant method which was created for temperate case studies and which showed low performances for intermittent rivers Ppastor et al. 2014). Moreover, an explanation on the choice of method (Tennant over Smakhtin, parametric vs. non-parametric methods) is required and why the latest methods were ignored (Hoekstra and Mekonnen 2011, Pastor, Ludwig et al. 2014).

I recommend the authors to test their parameters and methods with different case studies and other global global EF methods worldwide before validation and global upscaling. It is also necessary to compare these last EF methods in different contexts (different ecoregions, flow regime types). Overall, this study has the merit to extend knowledge on the eco-hydrology field but to my point of view it should be first described the use of NPP, resilience and trophic levels for all freshwater ecoregions (Abell, Thieme et al. 2008) including the acknowledgement of the latest global eco-hydrological studies (Oberdorff, Tedesco et al. 2011, Tisseuil, Cornu et al. 2013).

I wish good luck to the authors to develop further their study,

References

Abell, R., et al. (2008). "Freshwater ecoregions of the world: a new map of biogeographic units for freshwater biodiversity conservation." BioScience 58(5): 403-414.

Boulay, A.-M., et al. (2015). "Consensus building on the development of a stress-based indicator for LCA-based impact assessment of water consumption: outcome of the expert workshops." The International Journal of Life Cycle Assessment: 1-7.

Gaupp, F., et al. (2015). "The role of storage capacity in coping with intra-and inter-

annual water variability in large river basins." Environmental Research Letters 10(12): 125001.

Gerten, D., et al. (2013). "Towards a revised planetary boundary for consumptive fresh-water use: role of environmental flow requirements." Current Opinion in Environmental Sustainability 5(6): 551-558.

Hoekstra, A. Y. and M. M. Mekonnen (2011). Global water scarcity: the monthly blue water footprint compared to blue water availability for the world's major river basins. 333. 222. Delft, The Netherlands, , UNESCO-IHE Institute for Water Education. Value of Water Research Report: 78pp.

Oberdorff, T., et al. (2011). "Global and Regional Patterns in Riverine Fish Species Richness: A Review." International Journal Ecology 2011, Article ID 967631: 12 pp.

Pastor, A. V., et al. (2014). "Accounting for environmental flow requirements in global water assessments." Hydrol. Earth Syst. Sci. 18(12): 5041-5059.

Sadoff, C. W. (2015). Securing Water, Sustaining Growth: Report of the GWP/OECD Task Force on Water Security and Sustainable Growth.

Steffen, W., et al. (2015). "Planetary boundaries: Guiding human development on a changing planet." Science 347(6223): 1259855.

Tisseuil, C., et al. (2013). "Global diversity patterns and cross‐taxa convergence in freshwater systems." Journal of Animal Ecology 82(2): 365-376.

———————————————————

---

## Author Comment (AC1) · 13 Feb 2017

We greatly appreciate the reviewer's insightful comments on our paper. These comments have significantly helped us improve the manuscript.
We would like to answer each comment and to express the direction for revising the paper.

| | Referee's Comments | Authors' Answers |
|---|---|---|
| 1 | My main concern is that the authors have misconstrued the meaning of the incremental levels of flow identified in the Tennant method. This is critical because the selection of the low flow minimum and range (x1 in their calculation) exerts a dominant control on the eventual EFR. CDE and ERT simply adjust low flow recommendations within the range established by Tennant. The authors have interpreted that the incremental levels reflect variable ecological structure (i.e. increasing number of trophic levels), when in fact the levels reflect the condition (from natural to increasingly degraded) of the river ecological structure and function, whatever the natural state might be. This follows from a simplified view of the natural flow paradigm that the risk of degraded ecological condition increases as anthropic flow alteration increases. Instead, the authors assume that the incremental levels relate to the natural levels of ecological structure. The implication is that, in the author's approach, systems which have increasingly simpler ecological structures are expected to maintain equivalent ecological condition (relative to natural) with increasing anthropic flow alteration. I know of no ecological theory or research to support this assumption. If the authors are to continue with this approach they should present a clear theoretical justification and supporting research results. | As you indicated, the threshold of 10%, 30% and 60% of MAD in the Tennant method is a management level which focus on more or less artificially modified rivers and these levels are not correspond to ecological structures. The incremental levels in our model are independent from the Tennant's threshold. The authors assume that flow relates to ecological structure. That is, simpler ecological structure is expected to tolerate more flow reduction These assumptions will be supported by several classical theories and previous studies. The authors are well aware of this point, however, our explanation in the paper may be misleading and should correctly be described with corresponding references;

Our model which assess the structure (TI and vulnerability) from primary productivity will be supported by species-area theory (Macarthur and Wilson 1967). Species-energy theory advocates positive correlation between species richness and energy available, in turn, primary productivity in the area. This theory has also been supported by studies for riverine ecosystems. Oberdorff et al. (1995), and Guegan et al. (1998) investigated that the NPP is a surrogate for fish diversity in rivers. Species-area theory implies that species richness increases as surface area. The explanations for that are: larger area has lower extinction rate, higher speciation rate and higher habitat diversity (Hugueny et al. 2010). In regard to this theory, Guegan et al. (1998) showed that the total surface area of the river and the mean flow are the dominant factors for fish richness. Based on these ideas, in our study, we relate the amount of flow and habitat size, and suppose the rate of flow reduction that the target ecosystems can tolerate is different according to the TI. In addition to the species-area theory, the following facts may reinforce the correlation between flow and TI. For instance, large fish (which corresponds to the species of TI=4) most directly affected by flow reduction in the consequence of habitat reduction or disappearance, because large predators need larger territory for their life history and daily predation (Bunn 2002). Also large predators avoid shallow areas in order to hide themselves from birds and terrestrial predators (Creed 1990, Power 1995). The minimum threshold level in our model is set according to ecological structures expressed by TI. Our model which assess the structure (TI and vulnerability) from primary productivity will be supported by two classical theories in ecological richness: Species-energy theory (Wright 1983), |

[Figure]

The following figure shows the conceptual relations with tolerance of each TI and flow reduction. For the region of TI=4, if flow is reduced to Xb% of MAD, large fish at the top of the trophic level may difficult to survive. Therefore, Xa % of MAD is the minimum threshold level. Similarly, for the region of TI=3, in which small fish is the top of the trophic level, have relatively more tolerance against flow reduction, and minimum threshold level will be Xb % of MAD and so forth. On this minimum threshold level, additional rate will be added according to vulnerability (CDE and ERT).

There are no obvious threshold rates in ecological richness and flow, however, we set threshold rates as 60%, 30% and 10% of MAD in reference to the existing studies suggest environmental flow objectives.

*Reference:*

1. MacArthur, R. H., and E. O. Wilson.: The theory of island biogeography. Princeton University Press, Princeton, 1967.
2. Oberdorff, T., Guégan, J. F. and Hugueny, B.: Global scale patterns in freshwater fish species diversity. Ecography 18, 345–352, 1995.
3. Guégan, J.F., Lek, S. and Oberdorff, T.: Energy availability and habitat heterogeneity predict global riverine fish diversity. Nature (London) 391:382–384, 1998.
4. Hugueny, B., Oberdorff, T. and Tedescco, P.: Community Ecology of River Fishes: A Large-Scale, American Fisheries Society Symposium, 73, 2010.
5. Bunn, S.E. and Arthington, A.H.: Basic principles and ecological consequences of altered flow regimes for aquatic biodiversity, Environmental Management, 30, 492–507, 2002.
6. Creed, R.P.: Direct and indirect effects of crayfish grazing in a stream community, Ecology, 75, 2091-2103, 1994.
7. Power, M.E., Sun, A., Parker, G., Dietrich, W.E. and Wootton, J.T.: Hydraulic food-chain models, Bioscience, 45, 159-167, 1995.
8. Wright, D. H.: Species–energy theory: an extension of species–area theory. Oikos,41, 496–506. 1983.

| | | |
|---|---|---|
| 2 | Their assumption also erases from the calculation of EFR the essential (societal based) process of setting objectives for ecosystem management, such as the requirement of achieving 'good' ecological status in all water bodies of the European Union. The levelsset by Smakhtin et al. (2004) considered environmental management objectives (as required by best practice in setting ERTs), but the present paper does not. I recommend that the authors take note of this omission of management objectives in their approach and consider ways to rectify it. | As you indicated, the practical EFR should be incorporated to the management objectives with social aspects. Tennant's threshold is combined with the management level (societal based). Our model, in contrast, set the EFR set by ecological tolerance focused on the potential productivity of fluvial ecosystems without any human impact. We have a perspective to combine this threshold with social management objective in our following research, for example: $EFR = Q \times A_1 \times A_2$ $= Q \times (\text{societal based objective}) \times (\text{ecological based objective})$ where $A_1$ is the Tennant's management level and $A_2$ is the ecological tolerance in our model. In this paper, $A_1$ is considered as 1.0 (natural), therefore management level is the highest. For instance, when Q=mean annual discharge (MAD), $A_1 = 0.3$ (management level =fair) and $A_2 = 0.6$ (TI=4), EFR will be 18% of MAD. The proposed EFR in the existing model, Sone River (18.9% of MAD for moderately modified status, Joshi et al, 2014) and downstream of Zab river (18% of MAD using hydrologic methods, Abdi et al, 2015) are correspond to this level. We will add the explanation in our paper. *Reference:* 1. Joshi, K.D., Jha, D.N., Alam, A., Srivastava, S.K., Kumar, V. and Sharma, A.P.: Environmental flow requirements of river Sone: impacts of low discharge on fisheries, Current Science, 107, 478-488, 2014. 2. Abdi, R. and Yasi, M.: Evaluation of environmental flow requirements using eco-hydrologic-hydraulic methods in perennial rivers, Water Science and Technology, 72, 354-363, 2015. |
| 3 | Pg. 2, line 13: parts of the text beginning "Early approaches aimed to define..." appear to be copied and pasted or slightly modified from Pahl-Wostl et al. 2013. Any copied and pasted or slightly modified text in the manuscript should be deleted. Citing the source of copied and pasted text is not sufficient. All text not contained in quotation marks must be original and attributable to the authors alone. | The sentence was rephrased as follows; Originally, the environmental flow objectives have been mainly focused on habitat suitability for representative fish species, however, many researchers are now regarded that it is not sufficient to evaluate complex fluvial ecosystems as a whole (Acreman and Ferguson, 2010; Shafroth et al.; 2010, Pahl-Wostl et al., 2013). |
| 4 | Pg. 3, line 9: "Stream flow has often been treated as the 'master variable' since it can be readily described by indices" This is not the reason stream flow is considered as the master variable. Revert to the original source (Power, Mary E., et al. "Hydraulic food-chain models." BioScience 45.3 (1995): 159-167) to clarify. | We appreciate for the information about the important research. According to the paper by Power et al. (1995) and other researchers, we revised the sentence as follows; Stream flow is the major determinant of physical habitat and thus, a major determinant of biotic components (Bunn and Arthington, 2002). Power et al (1995) developed a hydraulic-food chain model using causal linkages between hydraulic parameters (depth, |

| | | |
|---|---|---|
| | | velocity and width) and trophic dynamics. The flow rate, determines other hydraulic parameters, considered as the master variable for evaluating ecological features of a stream.

 *Reference:*
 1.  Bunn, S.E. and Arthington, A.H.: Basic principles and ecological consequences of altered flow regimes for aquatic biodiversity, Environ. Manag., 30, 492-507, 2002.
 2.  Power, M.E., Sun, A., Parker, G., Dietrich, W.E. and Wootton, J.T.: Hydraulic food-chain models, Bioscience, 45, 159-167, 1995. |
| 5 | Pg. 4, line 7: Tharme 2003 is not correct reference for IFIM. Check and correct alignment of methods and original sources throughout paper. | We referred original sources for IFIM and PHABSIM:
 1.  Bovee K.D.: A Guide to Stream Habitat Analysis using the Instream Flow Incremental Methodology. US Fish and Wildlife Service Biological Services Programme, Co-operative Instream Flow Service Group, Instream Flow Information Paper No. 12. FWS/OBS-82–26. 1982. |
| 6 | Pg. 4, line 25: this section begins with the repetition of points made above. In fact there is quite a bit of redundancy throughout the manuscript that should be removed. | Short summery at each paragraph will be removed and redundancies will be eliminated throughout the paper. |
| 7 | Pg. 5, lines 3 and 4: change PRC to RPC. | PRC was corrected to RPM (riverine productivity model) |
| 8 | Pg. 9, TI section: the calculation of TI using this approach is overly detailed for the global scale and approach of the model. I recommend seeking a much simplified approach, taking into consideration my main concerns expressed at the beginning of this review. | As you pointed out, to apply such criteria at a global scale, it is necessary to simplify the model without omitting a fundamental mechanisms of the system. To this end, the authors tried to establish the TI based on the classical but authorized basic concept: species-energy theory. The theory implies that the greater primary productivity may lead to higher trophic diversification. The purpose of the TI is to offer simple boundaries of flow- related ecological structures (as is shown in the figure of answer1) for setting environmental flow criteria.
 If we try to express real trophic levels of a complex fluvial ecosystem, we have to consider metabolic process at each trophic levels, species interactions, as well as regional differences in metabolic rates. However, in order to apply the model globally, we simplify the mechanism as possible and used single set of target species.
 The other reviewer also commented to the structure of TI. Please refer to the additional explanation we will post to the other reviewer. |

| 9 | Pg. 10, beginning line 9: as mentioned in the initial paragraphs of this review, the authors have misconstrued the purpose of Tennant's incremental levels of river ecological condition. Please review Tennant's paper carefully and represent accurately in this paper. | The thresholds proposed in this study were set independently from Tennant's incremental levels. To make this clear, we rewrite the paragraph. (Please refer to the answer No.1) |
|---|---|---|
| 10 | Pg. 10, line 20: the switch from ratios (FV 80-8000) to flow magnitudes (>8m3/s) is unexpected and unexplained here. Is it correct? | The threshold values in the paper are correct. We used the four factors to classify the hydro-climatic regions through conditional branch (The Table below). The sentence was unclear, so we explain all of these thresholds according to table. |

| type | MMD | FV | MaxMD | AMT |
|---|---|---|---|---|
| Extremely Arid | < 0.03 | - | - | - |
| Savanna | $\geqq$ 0.03 | > 1,000 | - | - |
| Monsoonal | $\geqq$ 0.03 | $\leqq$ 1,000, 80< | - | - |
| Wet-moderate | $\geqq$ 0.03 | $\leqq$ 80 | > 8 | - |
| Moderate | $\geqq$ 0.03 | $\leqq$ 80 | $\leqq$ 8 | > 0 |
| Spring spate | $\geqq$ 0.03 | $\leqq$ 80 | $\leqq$ 8 | $\leqq$ 0 |

MMD (Mean Monthly Discharge):$m^3$/s/100km$^2$
FV (Flow Variability): maximum monthly discharge/ minimum monthly discharge
MaxMD (Maximum Monthly Discharge): $m^3$/s/100km$^2$
AMT(Average Monthly Temperature at coldest month): ℃

| 11 | Pg. 10, line 32: first mention of the Chikugo model for quantification of NPP. This is at the root of all following calculations but is not well described. First, the indication throughout the paper is that fluvial NPP is being calculated, but from what I read in Seino and Uchijima 1993 (not 2010), the model calculates terrestrial (or generic) NPP. How is 'fluvial' NPP calculated? If the model is not 'fluvial' specific there should be an explanation of the rationale the authors use for the model. Also, the model is described as "well-established" (line 32) but according to Google Scholar Seino and Uchijima (1993) has been cited only 5 times in 26 years. What is the rationale for "well-established" | We will add the following explanation in our paper;
There are two ways to get NPP: to calculate by a model, or to get measured values. Measured values are available from such as NASA, however, the advantage of using a model is that is able to calculate NPP under a variety of climatic conditions, for example using the data of future climate.
We have a perspective to simulate environmental flow under several climatic patterns, thus, Chikugo model is useful because it can calculate global NPP from basic climatic information. That is the reason why we used Chikugo model.

As you pointed out, the Chikugo model calculates terrestrial NPP. Besides solar radiation, NPP in a river is affected by other physical and chemical factors such as water temperature, nutrient concentration and turbidities (Woodward 2009). As far as authors know, none of the model to calculate fluvial NPP available and thus we applied terrestrial NPP in this model. Of course the terrestrial NPP does not completely correspond with the fluvial NPP, however, previous studies have been indicated that terrestrial and aquatic NPP co-vary closely (Livingstone et al., 1982, Oberdorff et al., 1995) mainly because aquatic plant production depends on the same latitudinal factors as terrestrial primary productivity. Using estimates of terrestrial NPP probably does not |

| | | |
|---|---|---|
| | | underestimate the energy available for riverine ecosystems (Hugueny et al., 2010) Some previous researches have been applied terrestrial NPP to assess the aquatic fish richness, since freshwater NPP was not available at a global scale (Oberdorff et al., 1995, Guegan et al. 1998).

The purpose of our study is not to reproduce the complex fluvial ecosystems, but to highlight regional characteristics under the same evaluation process. To this end, we regard Chikugo model as the most appropriate model available so far to estimate primary productivity. (We will add the reason for choosing Chikugo Model instead of using "well established").

*Reference:*
1. Woodward, G.:  Biodiversity, ecosystem functioning and food webs in fresh waters: assembling the jigsaw puzzle, Freshwater Biology, 54, 2171-2187, 2009.
2. Livingstone, D. A., M. Rowland, and P. E. Bailey.: On the size of African riverine fish faunas, American Zoologist 22, 361–369, 1982.
3. Oberdorff, T., Guégan, J. F. and Hugueny, B.: Global scale patterns in freshwater fish species diversity. Ecography 18, 345–352, 1995.
4. Guégan, J.F., Lek, S. and Oberdorff, T.: Energy availability and habitat heterogeneity predict global riverine fish diversity. Nature (London) 391:382–384, 1998.
5. Hugueny, B., Oberdorff, T. and Tedescco, P.: Community Ecology of River Fishes: A Large-Scale, American Fisheries Society Symposium, 73, 2010. |
| 12 | Pg. 11, line 30: _ is set as 3 globally, and length of grid cell is also the same, Therefore, it seems length is removed as a variable globally. Are there consequences to this simplification? | The length of the grid cell is different in latitudinal direction. It results in the difference in channel length.
When the parameter $\alpha$ changes from 2 to 4 (see pg.11 line 30), it does not show a linear increase since the flow velocity of each cell is different. Thus, when applying the single parameter ($\alpha$=3 as an average) , the calculated biomass will be slightly overestimated. |
| 13 | Pg. 13, line 12: tributaries and lakes are indicated as significant in influencing the results, but I do not understand how these are resolved (made significant) in the model. Are these resolved somehow independently in the 0.5x0.5 degree grid? If so explain. | Confluences of tributaries which are identified on the 0.5x0.5 gridded cells are considered here. In the river channel network model applied in this study does not actually distinguish confluences and lakes. Both of them are expressed as a grid to which two or more upstream cells are connected. However, we used "lakes" where we obviously identify the large lake on the 0.5x0.5 gridded model.
Confluences play as a biomass pool for downstream cells because of the following reason; If the catchment area is the same, biomass accumulation rate at the confluence cell is faster than that of without confluence, into which upstream biomass comes down step by step and certain amount of biomass dissipated at each cell. |

| 14 | Pg. 13, line 30: use of language like "this may be because. . ." suggests that the processes and relationships controlling ERT are not fully understood, but they are exactly known as represented in the mathematics of the model. Refer to the relationships in the model and explain more confidently. | We carefully examine the calculation process and figured out thet the length of the upstream reaches is the dominant factor for longer ERT. Therefore, we rephrased sentence "This is because..." and added the new explanation. If the CDE is the same rate, ERT is longer where the length of upper reaches is longer more dependent on biomass transported from upstream ($B_u$). For instance, at the middle of Ebro River, ERT is 56 and $B_u$ is 13% of total biomass, while at downstream of Parana River, where length of upper reaches is about 6-fold longer than Ebro, ERT is 180 and $B_u$ accounts for 60%. The latter case, more than half of the biomass originate from allochothonous, however, as only a small proportion of $B_u$ is transported downstream at each time steps across long distances, ERT becomes longer. |
|---|---|---|
| 15 | Pg. 15, line 1: "These regions are characterised as having low resilience, resulting in longer ERT. . ." What is the research evidence (papers?) for the lower resilience of large rivers outside of monsoonal regions and savanna regions? The authors have defined resilience based on ERT. Do not turn this around and assume low resilience because of the ERT calculated. | To state the casual relationship correctly, we rephrased the sentence; These regions are characterized as longer ERT, resulting in lower resilience. |
| 16 | Pg. 15, line 9: provide citations supporting that Smakhtin et al, results have been "widely applied in water resource assessments". | We rephrased and added the supporting references. The model of Smakhtin et al (2004) offers a first estimation the water required for the maintenance of freshwater ecosystems at the global scale. Their estimation have been referred by several global water recourse assessments (for example, Hanasaki et al., 2008, Rockstrom etal. 2009, Gleeson T. et al, Bonsch et al., 2015). *Reference:* 6. Hanasaki, N., Kanae, S., Oki, T., Masuda, K., Motoya, K., Shirakawa, N., Shen, Y. and Tanaka, K.: An integrated model for the assessment of global water resources –Part 2: Applications and assessments, Hydrol. Earth. Syst. Sci., 12, 1027-1037, 2008. 7. Rockstrom, J., Falkenmark, M., Karlberg, L., Hoff, H., Rost, S. and Gerten, D.: Future water availability for global food production: The potential of green water for increasing resilience to global change, Water. Resour. Res., 45, 2009. 8. Gleeson, T., Wada, Y., Bierkens, M.F.P., van Beek, L.P.H.: Water balance of global aquifers revealed by groundwater footprint, Nature, 488, 197-200, 2012. 9. Bonsch, M., Popp, A., Biewald, A., Rolinski, S., Schmitz, C., Weindl, I., Stevanovic, M., Hogner, K., Heinke, J. and Ostberg, S.: Environmental flow provision: Implications for agricultural water and land-use at the global scale, Global Environ. Chang., 30, 113-132 ,2015. |
| 17 | Pg. 15, line 23: Reference is made here to "feasible goals", which refer back to my concern about the lack of environmental management objectives in the | As is explained at the answer No.2, we suppose the management objective is the highest status (or natural), in order to highlight difference of ecological structures without any |

| | | |
|---|---|---|
| | approach of the authors. This needs more explicit attention in future versions of the model. | human impact. On the other hands, in the global assessment (Smakhtin et al 2004), EFR is assumed as a "fair" condition, in order to demonstrate a feasible management goal. Considering the management aspect, the equation (5) of Pg.9 should be expressed as follows.

$EFR = A_1 \times (x_1 + x_2) \times MAD$

Where $A_1$ is the management level. The $A_1$ will decide if the EFR is feasible in a management perspective. |
| 18 | Pg. 17, line 1: "We then improve the Tennant EFR. . ." as mentioned above I believe the authors have misconstrued the incremental levels of condition in Tennant, therefore I believe the Tennant method has been misused and not improved. Substantial attention is needed to address this in future versions of the model. | As you indicated, the expression of "improvement of Tennant method" is improper. It should be rephrased that we proposed a new principle of the thresholds focused on the ecological structure estimated by primary productivity which cannot be evaluated by flow regime only. In the future version of the model, the EFR should be combined with the threshold which has a management perspective, such as the method of Tennant. |

---

## Author Comment (AC3) · 14 Feb 2017

We greatly thank you for careful reading our manuscript and for giving important comments to improve our paper. We understand that the referee's main concerns are as follows; -The model is not based on the sufficient scientific evidences, especially it lacks the description about ecological metabolism.

-The model does not reflect recent empirical and theoretical contributions and rely only on some classic studies.

As the referee proposed, we started discussing with experts on freshwater ecology

and receiving advice about improving our model description. We try to add more science-based explanation according to relative literatures in the revised version of our manuscript. We appreciate the referee's concerns on the above mentioned point, however, we would like to express that the purpose of our study is not to evaluate a real tropic status of a river, but is to offer simple boundaries of flow-related ecological structures for setting environmental flow management criteria. As the referee pointed out, in order to express real trophic levels of a complex fluvial ecosystem, it is necessary to consider many factors such as metabolic process at each trophic level, species interactions in food webs, as well as regional differences in metabolic rates. To apply the criteria at a global scale at the resolution of 0.5x0.5 degrees, however, it is necessary to simplify the model without omitting fundamental mechanisms of the system. It is the reason that the authors' rely on few classical theories (Several exceptions and questions have been reported according to certain conditions, but these are not disproved.) For example, the authors tried to establish the TI, based on the classical species-energy theory (Wright 1983, Hugueny 2010). It advocates positive correlation between species richness and energy available, in turn, primary productivity in the area. This theory has also been supported by empirical studies for riverine ecosystems (Oberdorff et al. 1995, and Guegan et al. 1998).

In order to improve our manuscript, we will carefully explain the purpose and structures of our models with referring sufficient volume of recent literatures and knowledge.

Reference: 1. Wright, D. H.: Species–energy theory: an extension of species–area theory. Oikos,41, 496–506. 1983. 2. Hugueny, B., Oberdorff, T. and Tedescco, P.: Community Ecology of River Fishes: A Large-Scale, American Fisheries Society Symposium, 73, 2010. 3. Oberdorff, T., Guégan, J. F. and Hugueny, B.: Global scale patterns in freshwater fish species diversity. Ecography 18, 345–352, 1995. 4. Guégan, J.F., Lek, S. and Oberdorff, T.: Energy availability and habitat heterogeneity predict global riverine fish diversity. Nature (London) 391:382–384, 1998.

---

## Author Comment (AC4) · 14 Feb 2017

We wish to express our appreciation for your constructive comments and advices on our paper. In our paper, we outlined a new idea to set EFR criteria according to primary productivity, ripple effect and resilience. Therefore we applied single set of parameters for demonstrate results for the present phase of work. We agree that the model further needs to be tested and compared with recent ecohydrological studies in order to make the model applicable to global assessment. The authors are now working on parameters' verification referring empirical studies at each climatic regions of the world, and

these results should be reflected to the improved version of our model. We appreciate your providing information on related literatures. We will carefully review these studies and take into account the ideas and achievement of latest global ecohydrological studies.

---

## Referee Comment (RC3) · Anonymous Referee #3 · 15 Feb 2017

The authors present a evaluation of fluvial ecosystem via measures of productivity, ripple effect, and resilience in an effort to develop environmental flow requirement (EFR). The authors use three indices contribution of downstream ecosystems, trophic level and ecological recovery time in the estimation of the environmental flows. The authors argue that a conceptual EFR model based on plant biomass is adequate to measure productivity, ripple effect, and resilience in the system. Aquatic plants are broadly defined as either as vegetation biomass growing in the river or free floating algae.

The paper reads well, though could use some light editing. My biggest concern with the

paper is the ability to validate the results presented in the paper. the authors introduce two indices but present no tables presenting the data used or any other explanation. The only tables are for the trophic index (with table 3 ) providing some numbers used.

Also, the paper ignores the impact of other stressors to the systems (nutrients, chemicals are some examples).

For this paper to be considered for publication, I would recommend that the authors present a few more reliable case studies to show the validity of the indices introduced and the overall model.

Please see these other papers in the area

Yarnell, Sarah M., et al. "Functional flows in modified riverscapes: Hydrographs, habitats and opportunities." BioScience 65.10 (2015): 963-972.

Rheinheimer, D. E., S. M. Yarnell, and J. H. Viers. "Hydropower costs of environmental flows and climate warming in California's Upper Yuba River watershed." River Research and Applications 29.10 (2013): 1291-1305.

---

## Author Comment (AC5) · 15 Feb 2017

We thank the reviewer for his/her fruitful suggestions. We agree with the reviewer's recommendation that some case studies should be added to validate the indices. According to this, we will add the description of the model validation, and specify the presupposition of each index, including the reason of which factors have been taken into account and which have not (e.g. nutrients and chemicals). We also appreciate the reviewer's providing information on related literature. We will carefully read these studies and take them into account to improve our manuscript.